# Automatically Interpreting Millions of Features in Large Language Models

## Abstract

While the activations of neurons in deep neural networks usually do not have a simple human-understandable interpretation, sparse autoencoders (SAEs) can be used to transform these activations into a higher-dimensional latent space which can be more easily interpretable. However, SAEs can have millions of distinct latents, making it infeasible for humans to manually interpret each one. In this work, we build an open-source automated pipeline to generate and evaluate natural language interpretations for SAE latents using LLMs. We test our framework on SAEs of varying sizes, activation functions, and losses, trained on two different open-weight LLMs. We introduce five new techniques to score the quality of interpretations that are cheaper to run than the previous state of the art. One of these techniques, intervention scoring, evaluates the interpretability of the effects of intervening on a latent, which we find explains latents that are not recalled by existing methods. We propose guidelines for generating better interpretations that remain valid for a broader set of activating contexts, and discuss pitfalls with existing scoring techniques. Our code is available online.

## 1 Introduction

Large language models (LLMs) have reached human level performance in a broad range of domains (OpenAI, 2023), and can even be leveraged to develop agents (Wang et al., 2023) that can strategize (Bakhtin et al., 2022), cooperate and develop new ideas (Lu et al., 2024; Shaham et al., 2024). At the same time, we understand little about the internal representations driving their behavior. Early mechanistic interpretability research focused on analyzing the activation patterns of individual neurons (Olah et al., 2020; Gurnee et al., 2023; 2024). Due to the large number of neurons to be interpreted, automated approaches were proposed (Bills et al., 2023), where a second LLM is used to propose an interpretation given a set of neuron activations and the text snippets it activates on, in a process similar to that of generating a human label. But research has found that most neurons are "polysemantic", activating in contexts that can be significantly different (Arora et al., 2018; Elhage et al., 2022). The Linear Representation Hypothesis (Park et al., 2023) posits that human-interpretable concepts are encoded in *linear combinations* of neurons. A significant branch of current interpretability work focuses on extracting these latents and disentangling them (Bereska & Gavves, 2024).

Sparse autoencoders (SAEs) were proposed as a way to address polysemanticity (Cunningham et al., 2023). SAEs consist of two parts: an encoder that transforms activation vectors into a sparse, higher-dimensional latent space, and a decoder that projects the latents back into the original space. Both parts are trained jointly to minimize reconstruction error. SAE latents were found to be interpretable and potentially more monosemantic than neurons (Bricken et al., 2023; Cunningham et al., 2023). Recently, a significant effort was made to scale SAE training to larger models, like GPT-4 (Gao et al., 2024) and Claude 3 Sonnet (Templeton et al., 2024), and they have become an important interpretability tool for LLMs.

Training an SAE yields numerous sparse features, each of which needs a natural language interpretation. In this work, we introduce an automated framework that uses LLMs to generate an interpretation for each latent in an SAE. We use this framework to explain millions of latents across multiple models, layers, and SAE architectures. We also propose new ways to evaluate the quality of interpretations, and discuss the problems with existing approaches. We hope that these result can

inform other repositories of interpretations for SAE latents that already exist (Lin & Bloom, 2023), as well as improve comparisons between different SAE architectures.

## 2 RELATED WORK

One of the most successful approaches to automated interpretability focused on explaining neurons of GPT-2 using GPT-4 (Bills et al., 2023). GPT-4 was shown examples of contexts where a given neuron was active and was tasked to provide a short interpretation that could capture the activation patterns. To evaluate if a given interpretation captured the behavior of the neuron, GPT-4 was tasked to predict the activations of the neuron in a given context having access to that interpretation. The interpretation is then scored by how much the simulated activations correlate with the true activations. A similar approach was used in Templeton et al. (2024) to explain SAE latents of Claude 3 Sonnet. In general, current approaches focus on collecting contexts together with latent activations from the model to be explained, and use a larger model to find patterns in activating contexts.

Following those works, other methods of evaluating interpretations have been proposed, including asking a model to generate activating examples and measuring how much a "module" activates (Singh et al., 2023; Kopf et al., 2024). More recently, "interpretability agents" have been built, managing to do iterative experiment to find the best interpretations of vision neurons (Shaham et al., 2024). Potentially cheaper versions of automated interpretability have been suggested, where the model that is being explained doubles as an interpretation generation model (Kharlapenko et al., 2024). A prompt querying the meaning of a single placeholder token is passed to the model and latent activations are patched into its residual stream at the position of the placeholder token during execution, generating continuations related to the latent. This technique is inspired by earlier work on Patchscopes (Ghandeharioun et al., 2024) and SelfIE (Chen et al., 2024).

## 3 AUTOMATED INTERPRETABILITY PIPELINE

In this Section we explain, step-by-step, the main pipeline used to generate interpretations and score them. The pipeline can be broadly divided into 3 sequential steps. First, the activations of the SAEs to be interpreted are collected over a broad dataset. Then, for each latent, relevant contexts are selected and shown to an LLM which generates an interpretation for the activation pattern observed. Finally, these interpretations are matched with different contexts and used by an LLM in different tasks that evaluate how good the interpretations are in predicting activating and non-activating contexts, details explained below. As an illustration, we represent how that pipeline might look like for a real latent, see Figure 1.

### 3.1 COLLECTING ACTIVATIONS

Sparse autoencoders (SAEs) are feedforward networks with one hidden layer trained to reproduce their input using a sparse number of neurons. This means that while there are several times more neurons than there are dimensions of the input, only a small fraction of these neurons– less than 60 in this work– are non-zero and contribute to the output. In our pipeline, we seek to explain the non-zero activations of the SAE hidden layer neurons, which we call "latents."

We collected latent activations from the SAEs over a 10M token sample of RedPajama-v2 (RPJv2; Computer 2023). This is the same dataset that we used to train the Llama 3.1 8b SAEs, and consists of a data mix similar to the Llama 1 pretraining corpus.

We collected batches of 256 tokens starting with the beginning of sentence token (BOS).[1] The contexts used for activation collection are smaller than the contexts used to train the SAEs, and we find that on average, 30% of the latents of the 131k latent, per layer, Gemma 2 9b SAE don't activate more than 200 times over these 10M tokens and 15% don't activate at all. When we consider the training context length of 1024, only 5% of latents don't fire. When using a closer proxy of the training data, the "un-copyrighted" Pile, we find that the number of latents that activate fewer than

---

[1]We throw out the activations on the BOS token when generating interpretations for Gemma, as we were told in personal communication that these SAEs were not trained on BOS activations.

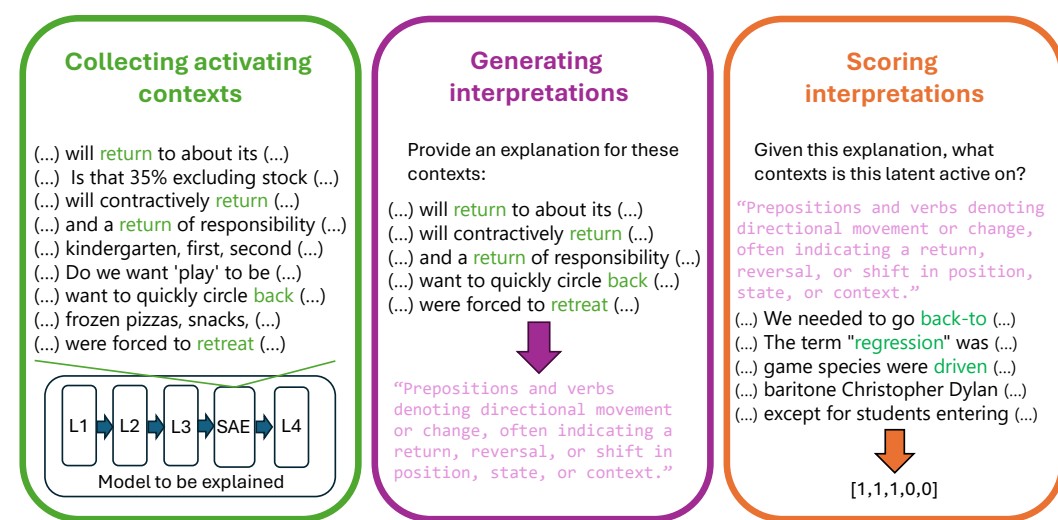

Figure 1: **Auto-interpretability pipeline.** The first step of the interpretability pipeline is collecting the activations of the target SAE over a broad range of text. In this figure we show short contexts for legibility, and over represent the active latent, which in reality would be active in a very small portion of the text. The activating contexts are selected and shown to an explainer LLM, prompt in Appendix A.2, which provides a short interpretation. This interpretation is then given to a scorer LLM that is tasked to use this interpretation to distinguish activating from non activating contexts, see Section 3.3

200 times decreases to 15%, and that only 1% don't activate at all, even when considering contexts of size 256. Interestingly, in the case of the 16k latent Gemma 2 9b SAE only around 10% of latents activate fewer than 200 times on the RPJv2 sample, suggesting that the larger 131k latent SAE learned more dataset specific latents than its smaller cousin.

## 3.2 GENERATING INTERPRETATIONS

Our approach to generating interpretations follows Bricken et al. (2023) in showing the explainer model examples sampled from different quantiles, but uses a more "natural" prompt where the activating example is shown whole, with the activating tokens emphasized and their activation strength shown after the example, see Appendix A.2 for the full prompt. For examples of some interpretations and their scores, see Figure A1.

We show the explainer model, Llama 3.1 70b Instruct, 40 different activating examples, each activating example consisting of 32 tokens, where the activating tokens can be on any position. While most latents are well-explained by contexts of length 32, we expect latents active over longer contexts to not be well captured by our pipeline. Using longer contexts to generate explanations would require either showing the model less examples, or using a much more powerful (and expensive) explainer model. Because we are able to evaluate interpretations significantly cheaper, bad interpretation can easily be filtered out, and we leave the explanation of "long-range" latents for future work. Using this interpretation setup would cost $200 to interpret 1 million latents.

We find that randomly sampling from a broader set of examples leads to interpretations that cover a more diverse set of activating examples, sometimes to the detriment of the top activating examples, see Figure 3. Using only top activating examples often yields more concise and specific interpretations that accurately describe these examples, but which fail to capture the whole distribution. On the other hand, stratified sampling from the deciles of the activation distribution can lead to interpretations that are too broad and that fail to capture a meaningful interpretation. For examples of these types of failure modes, see Figure A2 and the discussion in the Appendix A.3.

### 3.2.1 AUTOMATICALLY INTERPRETING INTERVENTIONS

Automatic interpretability methods, including most of those explored in this paper, typically look for correlations between activation of a latent and some natural-language property of the input. However, some latents are more closely related to what the model will output. For example, we find a latent[2] whose activation causes the model to output words associated with reputation but does not have a simple interpretation in terms of inputs.

We define an *output latent* as a latent that causes a property of the model's output that can be easily explained in natural language. See Section 3.3.5 for the definition we use in scoring.

Output latents can also be described in terms of their correlation with inputs. For example, the "reputation" latent activates in contexts where likely next tokens relate to reputation. However, explaining output latents in terms of causal influence on output has two advantages.

1. **Scalability**. Output latents are easier to describe in terms of effect on output because the explainer only needs to notice a simple pattern of output. The pattern of inputs this latent correlates with is more complex. Explaining a latent by correlating it with inputs requires approximating the computation of the subject model leading up to that latent, which may be challenging when subject models are highly capable and performing difficult tasks. Some latents' influence on output, however, might remain easily explainable.

2. **Causal evidence**. We might like to know that a latent causes a property of the model's output so that we can steer the model. Previous work has shown that existing auto-interpretability scores fail to accurately capture how well a given interpretation can predict the effect of intervening on a given neuron (Huang et al., 2023). Further, prior work has argued that causal evidence is more robust to distribution shifts (Bühlmann, 2018; Schölkopf et al., 2012).

### 3.3 SCORING INTERPRETATIONS

Not only should the generation of interpretations efficient, but so should the scoring of such interpretations. Practitioners would like to know how faithful each interpretation is to the actual behavior of the network. Some latents may not be amenable to a simple interpretation, and in these cases we expect the auto-interpretability pipeline to output an interpretation with poor evaluation metrics. Secondly, we can use evaluations as a feedback signal for the training of explainer models and tuning hyperparameters in the pipeline. Finally, some SAEs may simply be poor quality overall, and the aggregate evaluation metrics of the SAE's interpretations can be used to detect this.

The quality of SAE interpretations has so far been measured via simulation scoring, a technique introduced by Bills et al. (2023) for evaluating interpretations of neurons. It involves asking an explainer model to "simulate" a latent in a set of contexts, which means predicting how strongly the latent should activate at each token in a context given an interpretation. The Pearson correlation between the simulated and true activations is the simulation score. The standard in the literature is to sample contexts from a "top-and-random" distribution that mixes maximally activating contexts and contexts sampled uniformly at random from the activating corpus. While oversampling the top activating contexts introduces bias, it is used as a cheap variance reduction technique, given that simulation scoring over hundreds of examples per latent would be expensive (Cunningham et al., 2023).

In this work, we take a slightly different view of what makes for a good interpretation: an interpretation should serve as a *binary classifier* distinguishing activating from non-activating contexts. The reasoning behind this is simple. Given the highly sparse nature of SAE latents, most of their variance could be captured by a binary predictor that predicts the mean nonzero activation when the latent is expected to be active, and zero otherwise. In statistics, zero-inflated phenomena are often modeled as a mixture distribution with two components: a Dirac delta on zero, and another distribution (e.g. Poisson) for nonzero values. This goes against the general wisdom of simulation scoring, which focuses only on activating examples, even though they are less than 0.01% of the relevant contexts

---

[2]latent 157 of Gemma 2 9b's layer 32 autoencoder with 131k latents and an average L0 norm of 51. It has a detection score of 0.6.

for each latent of an SAE and that being able to distinguish non-activating contexts from activating contexts seems to be relatively more important.

## Detection

**Explanation:**
Words related to American football positions, specifically the tight end position
**Sentences:**
1: Patriots **tight end** Rob Gronkowski had his boss
2: names of months used in The Lord of the Rings
3: shown, is generally not eligible for ads. For example
**Correct output:**
[1,0,0]

## Fuzzing

**Explanation:**
Words related to American football positions, specifically the tight end position
**Sentences:**
1: Patriots <**tight end**> Rob Gronkowski had his boss
2: You should know this <about> **offensive line coaches**
3: <**running backs**>," he said. .. Defensive<**end**>
**Correct output:**
[1,0,1]

## Surprisal

**Explanation 1:**
Words related to American football positions, specifically the tight end position.
**Explanation 2:**
Sentences about dogs
**Sentence:**
Patriots **tight end** Rob Gronkowski had his boss -
**Correct output:**
P(Sentence|Exp 1)>P(Sentence|Exp 2)

## Embedding

**Explanation:**
Words related to American football positions, specifically the tight end position.
**Sentence 1:**
names of months used in The Lord of the Rings
**Sentence 2:**
Patriots **tight end** Rob Gronkowski had his boss -
**Correct output:**
Embed(S2)·Embed(Exp)>Embed(S1)·Embed(Exp)

Figure 2: **The new proposed scoring methods.** In **detection** scoring, the scorer model is tasked with selecting the set of sentences that activate a given latent given an interpretation. In this work, we show 5 examples at the same time, and each has an identical probability of being a sentence that activates the latent, independent of whether any other example also activates the latent. The activating tokens are colored in green for display, but that information is not shown to the scorer model. For **fuzzing** scoring, the scorer model is tasked with selecting the sentences where the highlighted tokens are the tokens that activate a target latent given an interpretation of that latent. In **surprisal** scoring, activating and non-activating examples are run through the model and the loss over those sentences is computed. Correct interpretations should decrease the loss in activating sentences compared to a generic interpretation, but shouldn't significantly decrease the loss in non-activating sentences. For **embedding** scoring, activating and non-activating sentences are embedded as "documents" that should be retrieved using the interpretation as a query.

As an alternative to simulation scoring, we introduce four new evaluation methods that focus on how well an interpretation enables a scorer to discriminate between activating and non-activating contexts. As an added benefit, all of these methods are more compute-efficient than simulation, see Table 1. We also propose a scoring method that evaluates the interpretability of interventions using a specific latent.

### 3.3.1 DETECTION

One simple approach for scoring interpretations is to ask a language model to identify whether a whole sequence activates a SAE latent given an interpretation. Detection requires few output tokens for each example used in scoring, meaning examples from a wider distribution of latent activation strengths can be used at approximately the same expense. By including non-activating contexts, this method measures both the precision and recall of the interpretation.

Detection is more "forgiving" than simulation insofar as the scorer does not need to localize the latent to a particular token. This means that detection evaluates whether an interpretation correctly identifies the types of contexts that a latent is active on, even if the interpretation does not correctly identify the actual token.

Both this method and fuzzing (described next) can leverage token probabilities to estimate how certain the scorer model is of their classification, and this is an effect that we believe can be to improve the scoring methods. Details on the prompt in Appendix A.4.1.

### 3.3.2 FUZZING

Fuzzing is similar to detection, but at the level of individual tokens. Here, potentially activating tokens are delimited in each example and the language model is prompted to identify which of the sentences are correctly marked. Fuzzing is the most similar with simulation. Because SAE activations are very sparse, simulation scoring mostly boils down to correctly identifying which of the tokens have non-zero activations. For this reason, fuzzing is the score that most correlates with simulation.

When interpretations focus on which tokens the latent activates, but not on which contexts, they can still score high on fuzzing, but comparatively lower on detection, specially if they are common tokens. Thus, evaluating an interpretation on both detection and fuzzing can identify whether a model is classifying examples for the correct reason. Details on the prompt on Appendix A.4.2.

### 3.3.3 SURPRISAL

Surprisal scoring is based on the idea that a good interpretation should help a base language model $\mathcal{M}$ (the "scorer") achieve lower cross-entropy loss on activating contexts than it would without a relevant interpretation. Specifically, for each context $\mathbf{x}$, activating or non-activating, we measure the *information value* of an interpretation $\mathbf{z}$ as $\log p_{\mathcal{M}}(\mathbf{x}|\mathbf{z}) - \log p_{\mathcal{M}}(\mathbf{x}|\tilde{\mathbf{z}})$, where $\tilde{\mathbf{z}}$ is a fixed pseudo-interpretation. A good interpretation should have higher information value on activating examples than on non-activating ones.

The overall surprisal score of the interpretation is given by the AUROC of its information value when viewed as a classifier distinguishing activating from non-activating contexts. Details on the prompt and how to compute the score in Appendix A.4.3.

Surprisal relies on the ability of the scorer model to tune its predictions based on the explanation, measuring both the relevance of the context given in the interpretation and the token specificity, but we believe that the current setup could be improved, as this is the score with the least correlation with the others.

### 3.3.4 EMBEDDING

Classifying between active and non-active contexts given a certain interpretation can also be seen as using interpretations of latents as "queries" that should be able to retrieve relevant "documents", contexts where the latent is active, between non-relevant "documents", non-activating contexts. This way, we take a selection of activating and non-activating contexts that embedded by an encoding transformer, and the similarity between the query and the documents is used as a classifier to distinguish between activating and non-activating contexts, and the score is given by the AUROC.

If the encoding model is small enough - we used a 400M parameter model - this technique is the fastest and opens up the possibility to evaluate a larger fraction of the activation distribution. We have seen that using a larger embedding model - 7B parameter model - didn't significantly improve the scores, see fig A3, although we believe that this approach was under-investigated. Details on the prompt, on the embedding model and on the way to compute the score in Appendix A.4.4.

### 3.3.5 INTERVENTION SCORING

Unlike the above four context-based scores, intervention scoring interprets a latent's counterfactual impact on model output. We quantify the interpretability of an intervention $I$ with interpretation $\mathbf{z}$ on a distribution of prompts $\pi$ as the average decrease in the scorer's surprisal about the interpretation when conditioned on text generated with the intervention.

$$S(I, \mathbf{z}; \pi) = \mathbb{E}_{\mathbf{x} \sim \pi}\left[\mathbb{E}_{\mathbf{i} \sim \mathcal{G}_I(\mathbf{x})}[\log p_{\mathcal{M}}(\mathbf{z}|\mathbf{i})] - \mathbb{E}_{\mathbf{g} \sim \mathcal{G}(\mathbf{x})}[\log p_{\mathcal{M}}(\mathbf{z}|\mathbf{g})]\right] \qquad (1)$$

$\mathcal{G}(\mathbf{x})$ is the distribution over subject model generations given prompt $\mathbf{x}$ at temperature 1. In $\mathcal{G}_I(\mathbf{x})$, the intervention is applied to the subject model as it generates. In practice we estimate the quantity $S$ by sampling one clean and one intervened generation for each sampled of prompt.

Sufficiently strong interventions can be trivially interpretable by causing the model to deterministically output some logit distribution. Therefore, **interpretability scores of interventions should be**

**compared for interventions of a fixed strength**. We define the strength $\sigma$ of an arbitrary intervention $I$ as the average KL-divergence of the model's intervened logit distribution with reference to the model's clean output.

$$\sigma(I; \pi) = \mathbb{E}_{\mathbf{x} \sim \pi} \left[ D_{KL}\left(p_{\text{subject}}(\cdot|\mathbf{x}) \,||\, p_{\text{subject}, I}(\cdot|\mathbf{x})\right) \right] \tag{2}$$

See Appendix A.7 for details on the interpretation and scoring pipeline we use in our intervention experiments.

## 4 RESULTS

### 4.1 COMPARING SCORING METHODS

When wanting to evaluate the explanation over a larger number of examples, simpler methods than simulation scoring have to be used (Templeton et al., 2024). Simulation scoring is traditionally done only in activating examples, ignoring whether the proposed explanation correctly handles non-activating examples. Simulation scoring is normally done in a small amount of examples per latent due to its high cost, and this is a significant disadvantage that prompted us to investigate more efficient scoring methods.

In Table 1 we compare the amount of tokens used to score a single latent using 100 different contexts, for both fuzzing, detection and two different simulation methods. We can see that fuzzing and detection are at least 5x cheaper than simulation when using the all-at-once (AAO) method described in Bills et al. (2023), although this method requires access to the log-probabilites of the prompt tokens, which are not accessible from providers of state-of-the-art closed-source models. A researcher wanting to generate simulation scores on the fly, without access to a local model, would have to use the token-by-token (TBT) method, which is 30x more than expensive than both fuzzing and detection. For the same set of intepretations, fuzzing and detection scores given by Llama 70b and Claude Sonnet 3.5 have similar distributions. On the other hand, Claude Sonnet 3.5 produces to higher simulation scores on average than Llama 70b, see Table A8 meaning that we should probably compare the price of fuzzing and detection on Llama 70b to the price of simulation using Claude Sonnet 3.5.

Embedding scoring is even cheaper, requiring 4000 input tokens for 100 different contexts, which at an average of $0.13 per million tokens would only cost $50/100k latents.

Table 1: Estimated cost of the different scoring methods. Brackets represent tokens that can be cached, potentially saving costs, although we do not do these calculations here. The number of input and output tokens is computed considering 100 examples show, and using fewer examples would cost proportionally less.

|  | Input tokens | Output tokens | Cost ($/100k latents) Llama 70b | Cost ($/100k latents) Claude Sonnet 3.5 |
|---|---|---|---|---|
| Fuzzing | 19.6k (14.2k) | 249 | 676 | 6.2k |
| Detection | 17.0k (11.9k) | 240 | 588 | 5.5k |
| Simulation (AAO) | 104.9k (87.5k) | 5 | 3.6k | 31.5k |
| Simulation (TBT) | 496.9k (451k) | 46.7k | 18.7k | 219.1k |

We propose that latent interpretations should be evaluated using more than a single technique when possible, because as discussed in Section 3.3, each of the proposed methods have different failure modes, and a more rich scoring framework can address some of these flaws. Specifically, we find that when computing the correlation of scores computed on the same 800 explanations, we find that while there is a clear positive correlation, some scores disagree– see Appendix for more examples.

We find fuzzing to be the most correlated with detection, and this is due to fact that fuzzing is mostly measuring how well the explanation can help the model predict which tokens have non-zero activation. Due to the sparsity of SAE activations, simulation scoring is doing a very similar test. On the contrary, we find that simulation has a significantly lower correlation with detection, embedding and surprisal scoring, as they more accurately measure whether a context activates a given latent,

instead of which token is active in a given context. When comparing the simulation scores provided by Claude 3.5 Sonnet, all correlations have a slight increase, see A.4.6.

Due to their relative cheapness compared with simulation scoring, we propose that fuzzing and detection scoring can be used to estimate whether explanations correctly identify on which tokens a latent is most likely to be active, and in which types of contexts this happens. Cheaper methods like embedding can be used to quickly iterate on explanations or to broadly separate latents that have bad explanations from those with good explanations, so that they can be refined.

Table 2: Spearman correlation, computed over scores of 800 different latent interpretations. For details on how the scores are computed and for the Pearson correlation, see Appendix A.4.6

|  | Fuzzing | Detection | Simulation | Embedding | Surprisal |
|---|---|---|---|---|---|
| Fuzzing | 1 | 0.73 | 0.75 | 0.41 | 0.30 |
| Detection |  | 1 | 0.44 | 0.71 | 0.62 |
| Simulation |  |  | 1 | 0.28 | 0.15 |
| Embedding |  |  |  | 1 | 0.79 |
| Surprisal |  |  |  |  | 1 |

### 4.1.1 INTERVENTION SCORING

The previously discussed scoring methods are *correlational* in the sense that they measure whether an interpretation can be used to predict the activation values of a given latent, or distinguish between activating and non-activating examples. Intervention scoring proposes to measure how well a given interpretation can predict the effect of *interventions* on the corresponding latent. Here we compare correlational interpretations generated with our pipeline and scored with fuzzing, to a set of interventional interpretations scored with intervention scoring. Our hypothesis is that some latents will have low correlational scores because their behavior is better explained by their downstream effects than by the contexts where they are active.

In Figure 4, we see that there is a slight negative correlation between the fuzzing score and the intervention score, showing that on average, latents with low fuzzing score can be better explained using their downstream effects than latents with high fuzzing score. We also find that the distribution of intervention scores of interpretation on a trained SAE is significantly different from the distribution of an interpretation of a randomly initialized SAE or a real SAE with randomly assigned interpretations, supporting the validity of this scoring method.

### 4.2 COMPARING INTERPRETATION METHODS

We use a set of 500+ latents as a testbed to measure the effects of design choices and hyperparameters on interpretation quality. Each latent is scored using 100 activating and 100 non-activating examples. The activating examples are chosen via stratified sampling such that there are always 10 examples from each of the 10 deciles of the activation distribution. We evaluate interpretation quality using fuzzing, detection and embedding scores, as those were both quick to compute and easy to interpret. We expect this mix of scores reflect the extent to which the proposed interpretation is valid over both activating and non-activating examples.

We find that the interpretations generated by considering only the activating contexts are significantly different from those generated by selecting activating contexts from the whole distribution. Interpretations based on top examples have higher specificity, but lower sensitivity. In fact, we observe that the sensitivity depends on how the degree of activation of each example shown, and that this sensitivity decreases faster for interpretation based on top examples. Interpretations generated from top activating examples are more concise and specific interpretations but fail to capture the whole distribution, while explanations based on examples sampled from the whole distribution lead to interpretations that are too broad, see Appendix A.3. This effect would not be seen if the scoring were done on just the most activating examples, underscoring a problem with current auto-interpretability evaluations, which produce interpretations using top activating examples and evaluate them on a small subset of the activation distribution.

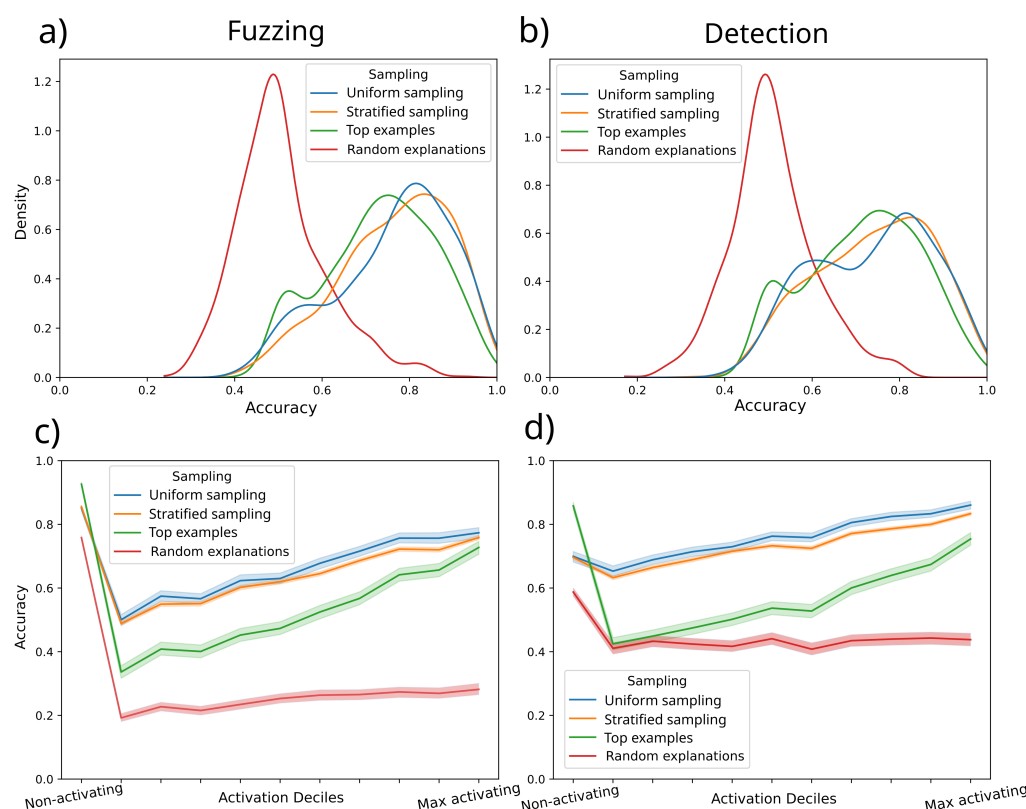

Figure 3: **Fuzzing and detection scores for different sampling techniques.** Panels **a)** and **b)** show the distributions of fuzzing and detection scores, respectively, as a function of different example sampling methods for interpretation generation. Sampling only from the top activation gets on average lower accuracy in fuzzing and on detection when compared with uniform sampling and stratified sampling. The distributions from random sampling and sampling from quantiles are very similar. Panels **c)** and **d)** measure how the interpretations generalize across activation quantiles, showing that interpretations generated from the top quantiles are better at distinguishing non-activating examples, but have lower accuracy on other quantiles, especially on the lower activating deciles. This also happens for the other interpretations, but the accuracy does not drop as much in lower activating deciles. We also show the scores of random explanations.

Increasing the size of the explainer model increases the scores of the interpretations, but we don't find the explanations generated by Claude Sonnet 3.5 to have much higher scores than those generated by Llama 3.1 70b, see A8, and both are similar to those generated by a human. Not surprisingly, we also see that even for simpler scores like fuzzing and detection, using a smaller scorer model leads to lower scores. It is possible that the explanations generated by Claude would be better than those generated by Llama 3.1 70b had we optimized the prompting techniques for that model.

Showing the explainer model a larger number of examples leads to slightly higher scores (Table A6). We find it doesn't matter much whether we use the same dataset as the training set of the SAEs (Table A2) or whether we slightly change the size of the contexts shown (Table A4). Using COT, at least when generating interpretations using Llama 70b, does not significantly increase their quality (Table A3) while significantly increasing the compute and time required to generate them. For this reason, we have not used it for our main experiments.

We find that SAEs with more latents have higher scores, and scores that are significantly higher than those of neurons. Neurons are more interpretable if made sparser by only considering the top $k$ most activated neurons on a given token, but still significantly underperform SAEs in our tests (Table A9). The location of the SAE matters; residual stream SAEs have slightly better scores than ones trained

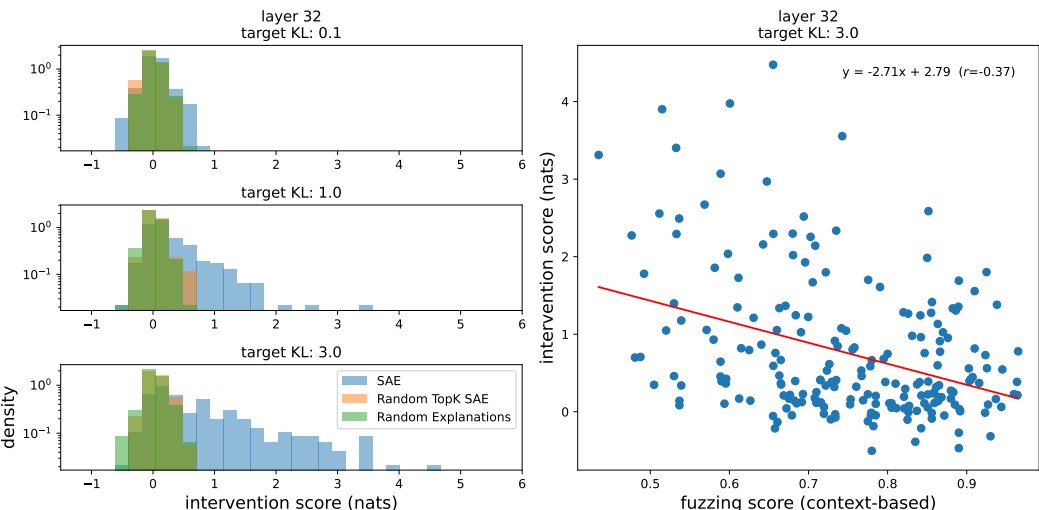

Figure 4: **Intervention scores.** Here we present intervention scores (Sec 3.3.5) for SAE latents in Gemma 2 9B at layer 32. **Left:** SAE latents are more interpretable than random latents, especially when intervening more strongly. Our explainer also produces interpretations that are scored higher than random interpretations. **Right:** Many latents that would normally be uninterpreted when using context-based automatic interpretability are interpretable in terms of their effects on output.

on MLP outputs (Table A9). We also observe that earlier layers have lower overall scores, but that the scores across model depth remain constant after the first few layers (Fig A5).

## 5 CONCLUSION

Explaining the latents of SAEs trained on cutting-edge LLMs is a computationally demanding task, requiring scalable methods to both generate interpretations and assess their quality. We addressed issues with the conventional simulation-based scoring and introduced five new scoring techniques, each with distinct strengths and limitations. These methods allowed us to explore the "prompt design space" for generating effective interpretations and propose practical guidelines.

Additionally, although current scoring methods do not account for interpretation length, we believe shorter interpretations are generally more useful and will incorporate this in future metrics. Some scoring methods also require further refinement, particularly in selecting non-activating examples to improve evaluation. We also investigated a method to generate and scoring interpretations of latents that is based on their downstream effect and shown that low score "correlational" scores could be due to the existence of "output" features.

Access to better, automatically generated interpretations could play a crucial role in areas like model steering, concept localization, and editing. We hope that our efficient scoring techniques will enable feedback loops to further enhance the quality of interpretations.

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

# A APPENDIX

## A.1 SAE INTERPRETATIONS OF A RANDOM SENTENCE

In this section, we show some non cherry-picked examples of latent interpretations on a sentence.

**Feature 6680, Layer 15 (0.90;0.70):** Third, second and first-person singular and plural pronouns, often used as subjects or objects in sentences, usually referring to people, animals or inanimate objects.

**Feature 3031, Layer 5 (0.95;0.88):** Third-person singular feminine pronouns and nouns indicating female presence.

**Feature 44, Layer 0 (0.68;0.56):** Proper nouns and common nouns that refer to people, places, ideas, or concepts, often denoting names, organizations, or entities, sometimes indicating presentations or resources, with varied applications and contexts.

**Feature 8603, Layer 5 (0.90;0.82):** A pronoun that signals the start of a quotation or dialogue.

"I spoke with her , and she had this awesome opportunity," he said. "And having the resources of the Center is awesome. It definitely outcompetes other programs I was interested in."

**Feature 4661, Layer 40 (0.63;0.59):** Verbs and verb-related phrases indicating necessity, ability, preparedness, or opportunity, often in a context of action, decision-making, or progress towards a goal.

**Feature 7343, Layer 20 (0.71,0.75):** Verb phrases or nouns representing actions, services, or objects related to online platforms, accounts, and transactions, often followed by specific details.

**Feature 2528, Layer 0 (0.70,0.57):** Various nouns denoting sports, social groups, and objects, often relating to concepts of identity, community, or competition.

**Feature 2011, Layer 0 (0.88;0.83):** "Adjectives describing a person's expertise or proficiency, often used to convey a sense of capability, such as \\experienced\\, \\skilled\\, \\knowledgeable\\, and \\qualified\\."

Figure A1: **SAE latents interpretations for a random sentence**. To visualize the latent interpretations produced, we select a sentence taken from the RPJv2 dataset. We selected 4 tokens in different positions in that sentence and filter for latents that are active in different layers. Then we randomly select active latents and their corresponding interpretations to display. We display the detection and fuzzing scores of each interpretation, which indicate how well it explains other examples in the dataset (see Section 3 for details on these scores). The latents selected had high activation, but were not cherry-picked based on interpretations or scores.

## A.2 EXPLAINER PROMPT

The system prompt if not using Chain of Thought (COT):

```
You are a meticulous AI researcher conducting an important
investigation into patterns found in language. Your task is to
analyze text and provide an interpretation that thoroughly
encapsulates possible patterns found in it.
Guidelines:

You will be given a list of text examples on which special words
are selected and between delimiters like << this >>.
If a sequence of consecutive tokens all are important,
the entire sequence of tokens will be contained between
delimiters <<just like this>>. How important each token is for
the behavior is listed after each example in parentheses.

- Try to produce a concise final description. Simply describe
the text latents that are common in the examples, and what
patterns you found.

- If the examples are uninformative, you don't need to mention
them. Don't focus on giving examples of important tokens,
but try to summarize the patterns found in the examples.
```

```
- Do not mention the marker tokens ($<<$ $>>$)
in your interpretation.

- Do not make lists of possible interpretations.
Keep your interpretations short and concise.

- The last line of your response must be the formatted
interpretation, using [interpretation]:
```

We add to the previous prompt the following if we want to do COT:

```
To better find the interpretation for the language patterns,
go through the following stages:

1.Find the special words that are selected in the examples
and list a couple of them. Search for patterns in these words,
if there are any. Don't list more than 5 words.

2. Write down general shared latents of the text examples.
This could be related to the full sentence or to the words
surrounding the marked words.

3. Formulate a hypothesis and write down the final interpretation
using [interpretation]:.
```

One of the few shot examples of how examples are displayed to the model.

```
Example 1:  and he was <<over the moon>> to find

Activations: (``over", 5), (`` the", 6), (`` moon", 9)

Example 2:  we'll be laughing <<till the cows come home>>! Pro

Activations: (``till", 5), (`` the", 5),
(`` cows", 8), (`` come", 8),
(`` home", 8)

Example 3:  thought Scotland was boring, but really there's
more <<than meets the eye>>! I'd

Activations: (``than", 5), (`` meets", 7), (`` the", 6),
(`` eye", 8)
```

If COT is used, an explicit example of using COT is demonstrated.

```
ACTIVATING TOKENS: ``over the moon", ``than meets the eye".
SURROUNDING TOKENS: No interesting patterns.

Step 1.
- The activating tokens are all parts of common idioms.
- The surrounding tokens have nothing in common.

Step 2.
- The examples contain common idioms.
- In some examples, the activating tokens are followed
by an exclamation mark.

Step 3.
```

```
– The activation values are the highest for the more common
idioms in examples 1 and 3.

Let me think carefully. Did I miss any patterns in the text
examples? Are there any more linguistic similarities?

– Yes, I missed one: The text examples all convey positive
sentiment.
```

Afterwards an interpretation is added to the example

```
[interpretation]: Common idioms in text conveying positive
sentiment.
```

### A.3 EXAMPLES OF ACTIVATING CONTEXTS AND INTERPRETATIONS

As discussed in the main text, the interpretations found for a given latent can be very different depending on the way to sample the activating contexts shown to the explainer model, see Figure A2. This has its advantages and disadvantages.

When using the top activating contexts, the explainer model normally gives an interpretation that is more narrow - "The concept of a buffer, referring to something that separates, shields, or protects one thing from another, often used in various contexts such as physical barriers, chemical reactions, or digital data processing" - instead of one that captures the full distribution - "Words or phrases associated with concepts of spatial or temporal separation (buffers, zones, or cushions) or colloquialisms (buff, buffs, or Buffy, referring to a popular TV show or enthusiast)". Here the narrower interpretations resulted in lower scores: the interpretation generated from examples sampled from the whole distribution scored 0.95 accuracy in fuzzing and 0.93 accuracy in detection, while the interpretation generated from top examples only achieves 0.8 accuracy in fuzzing and 0.74 accuracy in detection.

On the other hand, if we look at the second example, sampling from the whole distribution examples may sometimes confuse the explainer model - "Tokens often precede or succeed prepositions, articles, and words that signal possession or quantity, frequently indicating a relationship between objects or actions. of latent", which might not see the pattern that is more clear in the top activating examples -"Descriptions of food or events where food is involved, often mentioning leftovers, and sometimes mentioning the act of eating, serving, or storing food, as well as the amount of food, or the pleasure or satisfaction derived from it." Here we see very poor scores on the interpretation from randomly sampled examples, 0.54 accuracy in detection and 0.57 accuracy in fuzzing, while the interpretation generated from the top examples has 0.89 accuracy both in detection and fuzzing,

### A.4 DIFFERENT SCORING METHODS

#### A.4.1 DETECTION DETAILS

The prompt used for detection scoring is the following:

```
You are an intelligent and meticulous linguistics researcher.

You will be given a certain latent of text, such as
``male pronouns" or ``text with negative sentiment".

You will then be given several text examples. Your task
is to determine which examples possess the latent.

For each example in turn, return 1 if the sentence is
correctly labeled or 0 if the tokens are mislabeled. You must
return your response in a valid Python list. Do not return
anything else besides a Python list.
```

Together with the prompt, there are several few shot examples like the following:

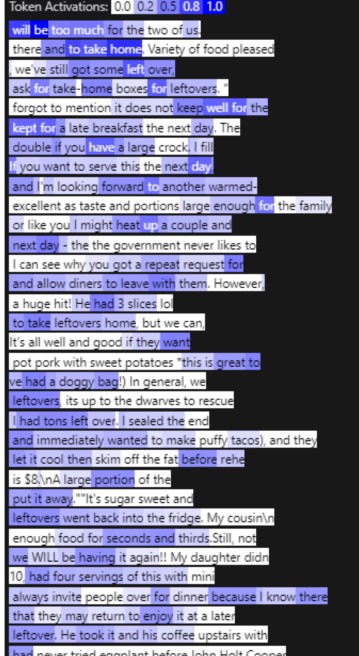

Figure A2: Activating contexts of latent 209 (top) and 293 (bottom), from layers 8 and 32, respectively, of the 131k latent SAE trained on the residual stream of Gemma 2 9b. The shown examples are similar to the ones given to the explainer model to come up with interpretations.

```
<user_prompt>

latent interpretation: Words related to American football
positions, specifically the tight end position.

Text examples:

Example 0:Getty Images Patriots tight end Rob Gronkowski
had his boss

Example 1: names of months used in The Lord of the Rings:
the

Example 2: Media Day 2015 LSU defensive end Isaiah Washington
(94) speaks to the

Example 3: shown, is generally not eligible for ads. For
example, videos about recent tragedies,

Example 4: line, with the left side namely tackle Byron
Bell at tackle and guard Amini

<assistant_response>

[1,0,0,0,1]
```

In this example the <user_prompt> and the <assistant_response> tags are substituted with the correct instruct format used by the scorer model. In detection scoring, 5 shuffled examples are shown to the model at the same time.

### A.4.2 FUZZING DETAILS

The prompt used for fuzzing scoring is the following:

```
You are an intelligent and meticulous linguistics researcher.

You will be given a certain latent of text, such as ``male
pronouns" or ``text with negative sentiment". You will
be given a few examples of text that contain this latent.
Portions of the sentence which strongly represent this
latent are between tokens << and >>.

Some examples might be mislabeled. Your task is to determine
if every single token within << and >> is correctly
labeled. Consider that all provided examples could be correct,
none of the examples could be correct, or a mix. An example
is only correct if every marked token is representative
of the latent

For each example in turn, return 1 if the sentence is correctly
labeled or 0 if the tokens are mislabeled. You must return
your response in a valid Python list. Do not return anything
else besides a Python list.
```

Followed by few-shot examples of the kind:

```
<user_prompt>
```

```
latent interpretation: Words related to American football
positions, specifically the tight end position.

Text examples:

Example 0:Getty Images Patriots<< tight end>> Rob Gronkowski
had his boss

Example 1: posted You should know this<< about>> offensive
line coaches: they are large, demanding<< men>>

Example 2: Media Day 2015 LSU<< defensive>> end Isaiah
Washington (94) speaks<< to the>>

Example 3:<< running backs>>,'' he said. .. Defensive
<< end>>
Carroll Phillips is improving and his injury is

Example 4:<< line>>, with the left side namely<< tackle>>
Byron Bell at<< tackle>> and<< guard>> Amini

<assistant_response>

[1,0,0,1,1]
```

In this example, the <user_prompt> and the <assistant_response> tags are substituted with the correct instruct format used by the scorer model. In fuzzing scoring, 5 shuffled examples are shown to the model at the same time.

### A.4.3 SURPRISAL DETAILS

In surprisal scoring, the cross-entropy loss of the scorer model is computed over the tokens of an example. For each interpretation, this loss is computed with the interpretation and with a default interpretation - "Various unrelated sentences," where the examples can either be activating context or non-activating contexts. In our first approach, Llama 3.1 70b base was used. The prompt starts with few shot examples like:

```
The following is a description of a certain latent of text
and a list of examples that contain the latent.

Description:

References to the Antichrist, the Apocalypse and conspiracy
theories related to those topics.

Sentences:

 '' by which he distinguishes Antichrist is, that he would
 rob God of his honour and take it to himself, he gives
 the leading latent which we ought  "

 ''3 begins. And the rise of Antichrist. Get ready with   "

 '' would be destroyed. The worlds economy would likely
 collapse as a result and could usher in a one world government
 movement. I wrote a small 6 page  "

Description:
```

```
Sentences containing digits forming a four-digit year.

Sentences:

  `` 20, 2013 at 7:41 pm Martin Smith  "

  `` of 2012. In other words, Italy's   "

  ``end 2012 levels). In the first quarter of 2013, we expect
  revenue to be up slightly from the fourth quarter  "

Description:

Text related to banking and financial institutions.

Sentences:

  ``: He is on the Board of Directors with the Lumbee Bank   "

  `` refurbishing the Bank's branches. BIP reached 400
  thousand users in one year The use of BIP has already
  doubled The  "

  `` the Federal Deposit Insurance Corp.  "

Description:

Occurrences of the word 'The' at the beginning of sentence.

Sentences:

  ``The Smoking Tire hits the canyons with one of the fastest
  Audi's on the road  "

  ``The Chairman of the ABI  "

  ``The administrative center is the town of Koch.  "
```

With the pairs of losses with the interpretation and with the default interpretation, it is possible to compute the decrease in loss caused by having access to the interpretation. It is expected that in activating contexts this difference will be greater than in non-activating examples, and the score of the interpretation is given by the AUC computed using this loss as a proxy for activating and non-activating labels.

### A.4.4 EMBEDDING DETAILS

For embedding scoring, we use a small 400M parameter model. We chose a small performant model on MTEB (Muennighoff et al., 2022) because we found similar scores when using a larger 7B parameter model, see fig A3, as this size allowed us to do increase the number of examples used in scoring.

A set of activating and randomly selected non-activating examples are embedded using the scorer model. Then a "query" instruction is embedded:

```
Instruct: Retrieve sentences that could be related to
the interpretation. Query:  \{interpretation\}
```

The cosine-similarity between the instruction embedding and the examples is computed and is used as a proxy for activating and non-activating labels when computing the AUC, which is the score of that interpretation.

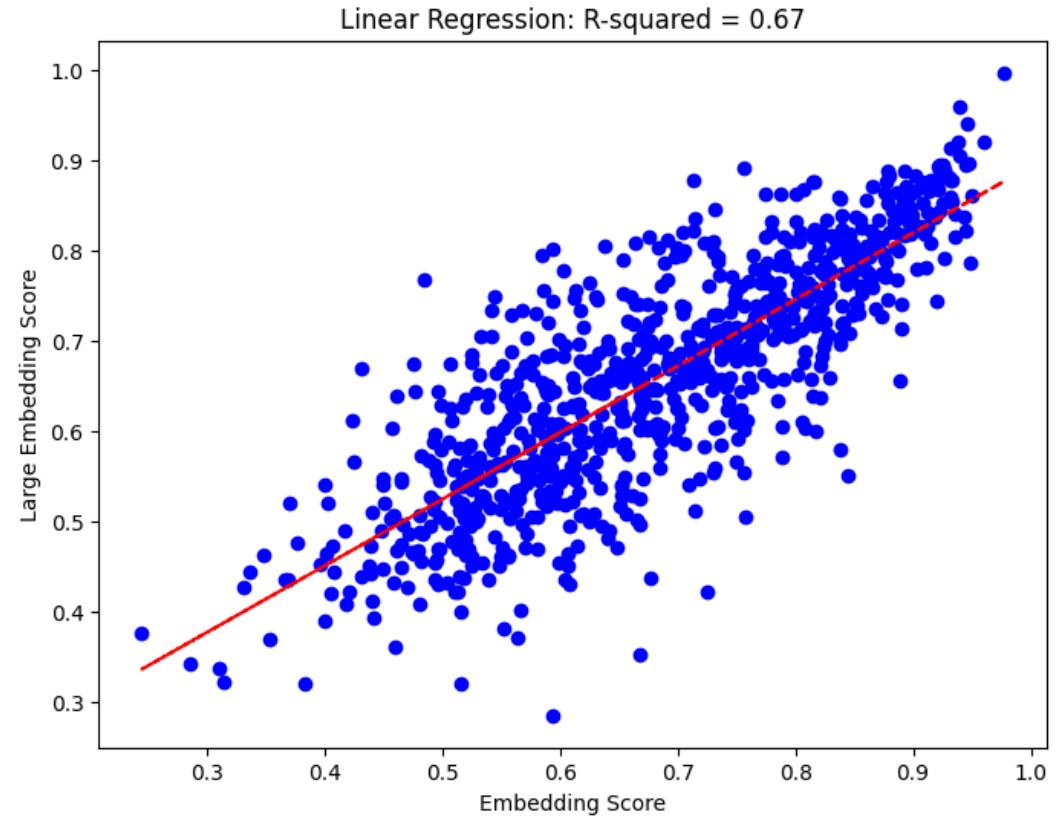

Figure A3: **Comparison between the scores given by a small embedding model and a larger one**

### A.4.5 ADVERSARIAL EXAMPLES

Most latents are active in less than 0.1% of the full dataset used to collect the activations, making random non-activating contexts very diverse. Randomly sampling non-activating examples cannot be used to determine whether a latent fires in a token in a specific context or on that token in general, as it is unlikely that that token randomly occurs in each non-activating example. As SAEs are scaled and latents become sparser and more specific, techniques that overly rely on activating contexts will have more imprecise results Gao et al. (2024).

Motivated by the phenomenon of latent splitting, we could use "similar" latents to test whether interpretations are precise enough to distinguish between similar contexts. A potential approach is using cosine similarity between decoder directions of latents to find counterexamples for an interpretation. Some works (Juang et al., 2024; Macgrath, 2024) have shown that a significant fraction of current latent interpretations can't be used to distinguish between latents with high similarity.

### A.4.6 CORRELATION BETWEEN SCORES

We compute the correlation between these different scores and simulation 800 different latent interpretations, of the 131k latent SAE trained on the residual stream of Gemma 2 9b spread across 4 different layers. We use 100 activating example, 10 from each of 10 different activating quantiles, and 100 non-activating examples to compute these scores and use Llama 3.1 70b Instruct as the scorer model for simulation, fuzzing and detections, while we use Llama 3.1 70b Base for surprisal and Stella-400M as the embedding model.

We also measure the correlation between our scoring methods and scores given by humans. Humans were tasked to score how an explanation related to a given context, similar to that done in (Templeton et al., 2024). The human evaluators could choose between 4 options, "1 - Explanation not related to

Table A1: Pearson correlation computed over 800 different latent scores

|  | Fuzzing | Detection | Simulation | Embedding | Surprisal |
|---|---|---|---|---|---|
| Fuzzing | 1 | 0.74 | 0.74 | 0.42 | 0.32 |
| Detection |  | 1 | 0.46 | 0.70 | 0.62 |
| Simulation |  |  | 1 | 0.30 | 0.14 |
| Embedding |  |  |  | 1 | 0.79 |
| Surprisal |  |  |  |  | 1 |

the text", "2 - Explanation vaguely related to the text", "3 - Incomplete explanation but present in the text", "4 - Explanation describes the activation of the tokens". We measure the average score of a latent in at least 5 contexts. In total 700 contexts and 81 latent interpretation. Because the sample size is smaller, we report the correlation separately from tables A1 and 2, which have significantly better statistics. We find that fuzzing as the best Spearman correlation with human scores - 0.69 - followed by simulation - 0.60 - and recall - 0.59. Surprisal and embedding have 0.34 and 0.32 Spearman correlation respectively.

Interestingly, we also find that the correlation between simulation scores and other scores also increases when the simulation scorer is Claude Sonnet 3.5. These estimations are on a smaller number of latents (100), due to the high cost of simulation using Claude Sonnet, but on this set, the spearman correlation between fuzzing and simulation increases from 0.69 to 0.75, between detection and simulation increases from 0.31 to 0.38, between embedding and simulation from 0.14 to 0.21 and from surprisal to simulation from 0.02 to 0.10. The baseline numbers for the correlation of simulation scores are lower on this set, so we expect that had we done Claude scores on a larger set of explanations, the correlations between the scores would be higher.

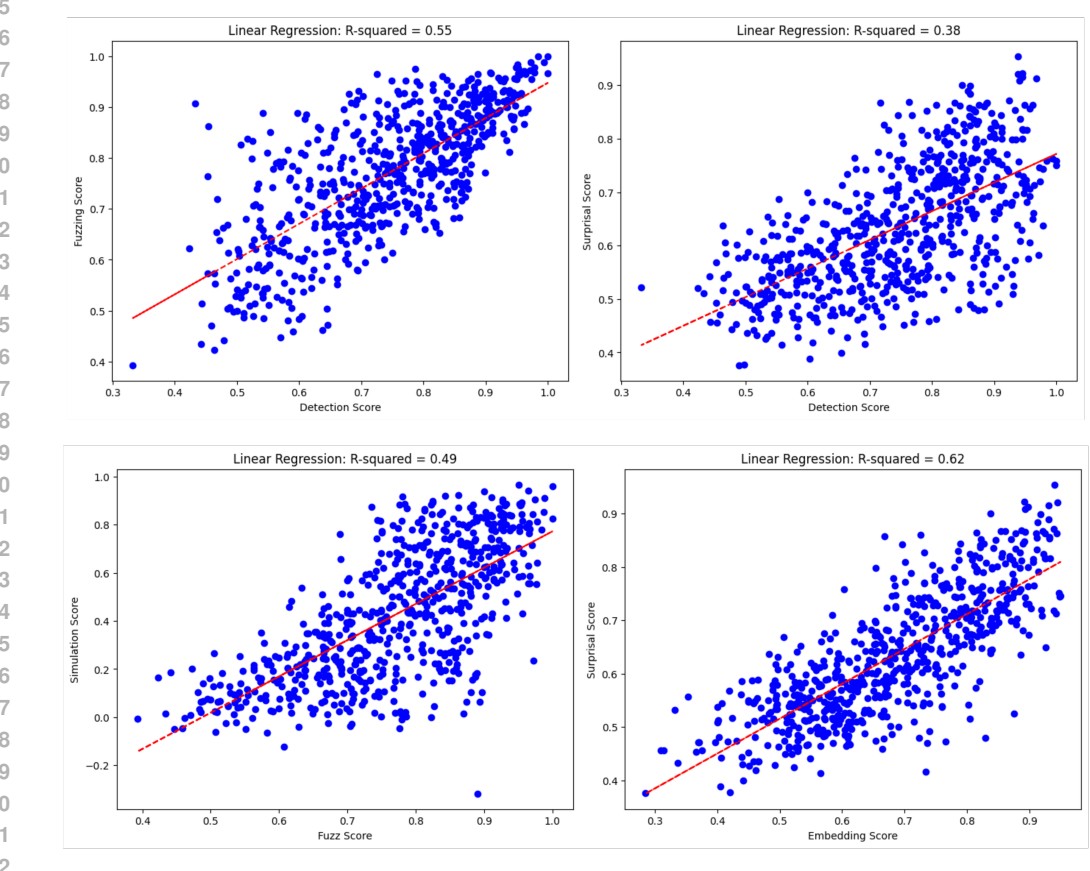

Figure A4: **Scatter plots with different combinations of scores**

### A.4.7 Disagreement between scores

In this section, we will discuss how different types of scores might evaluate explanations differently. In particular, we will focus on an interpretation with fuzzing score but low detection score, high fuzzing score but low simulation score, high simulation score but low embedding score and high embedding score and low simulation score.

For instance, feature 2 of layer 8 of the 131k latent SAE model trained on the residual stream of Gemma 2 9b has a fuzzing score of 0.9 but a detection score of only 0.43. The interpretation given by our pipeline is "Verbs that link a subject to additional information, often in a formal or descriptive tone." Most of the activations of this latent are on sentences like "This <is needed> because Magento (...)" or "For instance, this< is> technically correct syntax", where <> represent active tokens. In both these examples, the model correctly identifies the highlighted tokens as active during fuzzing, but incorrectly identifies the context as non-active during detection. On the other hand, detection incorrectly identifies sentences like "pay for things that would prevent larger issues down the road is better in the long run." or "Neuroscientist Jack Gallant calls the research a technologic tour de force and says the ultimate decoder would provide vivid" as active. This explanation is too vague on the activating context, leading to a low detection score, but specific enough in the types of tokens that are active, leading to a high fuzz score.

Feature 281 of the same layer has a fuzzing score of 0.97, but a simulation score of only 0.19. Its interpretation is "URLs or hyperlinks containing query parameters, indicating a request for specific data or actions on a web page." The simulation score is lower for than the fuzzing score because the model has to decide which parts of the URL the latent is active on, while the fuzzing scorer either is shown links that are highlighted which it will say are active or non-links that are highlighted and are, most likely, not active.

Feature 8 of layer 24 has a simulation score of 0.73, but an embedding score of only 0.43. Its interpretation is "Prevalence of logical operators and conjunctions in text, including simple addition and conjunction in various contexts, as well as indicators of contrasting or additional information, often used for comparison or to provide supplementary details". While doing simulation, the model correctly identifies that in the sentence "the cost of the unit itself, plus installation and construction costs.", the token "plus" is active but all the others aren't. The same is true for sentences like "Plus my potato bread I made on the ANZAC Day" and "Cliff Chiang on reintroducing Orion in Wonder Woman! Plus interviews". On the other hand, these sentences don't have high embedding similarity with the explanation.

Feature 10 of layer 16 has the opposite situation happening, with an embedding score of 0.74 and a simulation score of only 0.04. Its interpretation is "Verbs, prepositions, and adverbs that connect clauses or indicate direction, movement, or progression, often in a sequence of actions or events". In sentences like "right, three months ¡to close down¿. There was dead silence when the message was read. Everybody waited for Mr. Smith to speak. Mr. Gingham" the simulator model incorrectly identifies " waited" and the second " to" as active, and misses the real activation in "to close down". The same happens in "This is ¡going¿ to ¡be¿ one ¡swinging¿ party. Especially if the guests survive Uncle Wilhelm. I love weddings. I love going to them, I love being in", where the simulator model identified "love being in" as well as "love going to" as being active. The embedding model correctly identified that most activating sentences have the mention of motion.

With these examples, we hope to demonstrate that these generated explanations are not perfect, but that we have an easier understanding of their flaws by looking at which cases different scores disagree on.

### A.5 SAE models used

Throughout this work, we used different SAEs trained on Gemma. The 16k latent ones trained on the MLP have the following average L0 norms per layer:

```
0:50, 1:56, 2:33, 3:55, 4:66, 5:46, 6:46, 7:47, 8:55, 9:40, 10:49, 11:34, 12:42, 13:40, 14:41,
15:45, 16:37, 17:41, 18:36, 19:38, 20:41, 21:34, 22:34, 23:73, 24:32, 25:72, 26:57, 27:52,
28:50, 29:49, 30:51, 31:43, 32:44, 33:48, 34:47, 35:46, 36:47, 37:53, 38:45, 39:43, 40:37,
41:58
```

Table A2: Impact of training dataset on Fuzzing and Detection performance. Numbers shown are the median score and the interquartile (25%-75%) range.

| Experiment | Fuzzing | Detection | Embedding |
|---|---|---|---|
| Random interpretation | 0.51 (0.45–0.57) | 0.51 (0.45–0.58) | 0.51 (0.44–0.57) |
| Randomly initialized Topk SAE | 0.55 (0.50–0.60) | 0.54 (0.50–0.59) | – |
| RPJ-v2 | 0.76 (0.67–0.86) | 0.74 (0.63–0.85) | 0.67 (0.57–0.80) |
| Pile | **0.76 (0.67–0.86)** | **0.76 (0.67–0.85)** | **0.69 (0.57–0.80)** |

Table A3: Impact of prompt content on Fuzzing and Detection performance. Numbers shown are the median score and the interquartile (25%-75%) range.

| Experiment | Fuzzing | Detection | Embedding |
|---|---|---|---|
| Random interpretation | 0.51 (0.45–0.57) | 0.51 (0.45–0.58) | 0.51 (0.44–0.57) |
| Randomly initialized Topk SAE | 0.55 (0.50–0.60) | 0.54 (0.50–0.59) | – |
| Activations in prompt | **0.76 (0.67–0.86)** | **0.74 (0.63–0.85)** | **0.68 (0.57–0.80)** |
| No activations in prompt | 0.75 (0.65–0.86) | 0.73 (0.60–0.84) | 0.68 (0.58–0.79) |
| COT in prompt | 0.76 (0.67–0.86) | 0.73 (0.61–0.85) | 0.65 (0.55–0.75) |

The 16k latent ones trained on the residual stream have the following average L0 norms:

0:35, 1:69, 2:67, 3:37, 4:37, 5:37, 6:47, 7:46, 8:51, 9:51, 10:57, 11:32, 12:33, 13:34, 14:35, 15:34, 16:39, 17:38, 18:37, 19:35, 20: 36, 21:36, 22: 35, 23: 35, 24: 34, 25: 34, 26: 35, 27:36, 28: 37, 29:38, 30:37, 31:35, 32: 34, 33:34, 34:34, 35:34, 36:34, 37:34, 38:34, 39:34, 40:32, 41:52

The 131k latent ones trained in the residual steam have the following average L0 norms:

0:30, 1:33, 2:36, 3:46, 4:51, 5:51, 6:66, 7:38, 8:41, 9:42, 10:47, 11:49, 12:52, 13:30, 14:56, 15:55, 16:35, 17:35, 18:34, 19:32, 20:34, 21:33, 22:32, 23:32, 24: 55, 25:54, 26:32, 27:33, 28: 32, 29:33, 30:32, 31:52, 32: 51, 33:51, 34:51, 35:51, 36: 51, 37:53, 38:53, 39:54, 40: 49, 41:45

We also used SAEs with 65k latents, trained on the residual stream and the MLP of Llama 8b.

## A.6 FACTORS THAT INFLUENCE THE EXPLAINABILITY

### A.6.1 DEPENDENCE ON DATASET

Even though a significant portion of SAE latents are less active when using RPJv2 instead of the Pile, we find that the latents that are active are generally interpretable to the same degree. These evaluations were done using the 131k latent SAE trained on the residual stream of Gemma 2 9b. The scorer and the explainer model where Llama 3.1b 70b instruct, quantized to 4bit.

### A.6.2 DEPENDENCE ON CHAIN OF THOUGHT AND ACTIVATION INFORMATION

We find that COT slightly increases the scores of the interpretations found, and that it significantly slows down the rate at which one can produce interpretations. Giving the explainer model, the activations associated with each token seem to slightly increase the scores of the interpretations generated. These evaluations were done using the 131k latent SAE trained on the residual stream of Gemma 2 9b. The scorer and the explainer model where Llama 3.1b 70b instruct, quantized to 4bit.

### A.6.3 DEPENDENCE ON CONTEXT LENGTH

The context length of the shown examples did not significantly change the scores obtained for the interpretations. These evaluations were done using the 131k latent SAE trained on the residual stream of Gemma 2 9b. The scorer and the explainer model were Llama 3.1b 70b instruct, quantized to 4 bits.

Table A4: Impact of context length on Fuzzing and Detection performance. Numbers shown are the median score and the interquartile (25%-75%) range.

| Experiment | Fuzzing | Detection | Embedding |
|---|---|---|---|
| Random interpretation | 0.51 (0.45–0.57) | 0.51 (0.45–0.58) | 0.51 (0.44–0.57) |
| Randomly initialized Topk SAE | 0.55 (0.50–0.60) | 0.54 (0.50–0.59) | – |
| 16 context | 0.75 (0.65–0.86) | 0.74 (0.62–0.85) | 0.70 (0.59–0.81) |
| 32 context | **0.76 (0.67–0.86)** | **0.74 (0.63–0.85)** | 0.67 (0.57–0.80) |
| 64 context | 0.74 (0.64–0.64) | 0.70 (0.57–0.81) | 0.65 (0.54–0.78) |

Table A5: Impact of example sampling strategies on Fuzzing and Detection performance. Numbers shown are the median score and the interquartile (25%-75%) range.

| Experiment | Fuzzing | Detection | Embedding |
|---|---|---|---|
| Random interpretation | 0.51 (0.45–0.57) | 0.51 (0.45–0.58) | 0.51 (0.44–0.57) |
| Randomly initialized Topk SAE | 0.55 (0.50–0.60) | 0.54 (0.50–0.59) | – |
| Randomly sampled | 0.76 (0.68–0.86) | 0.74 (0.62–0.84) | 0.66 (0.56–0.78) |
| Sampled from quantiles | **0.77 (0.69–0.87)** | **0.74 (0.64–0.85)** | 0.68 (0.57–0.80) |
| Sampled from top examples | 0.73 (0.64–0.83) | 0.72 (0.62–0.82) | **0.70 (0.58–0.80)** |

### A.6.4  DEPENDENCE ON THE ORIGIN OF EXAMPLES

Sampling the examples from the whole distribution of activations, or sampling a fixed number of examples over different activation quantiles, significantly improved the scores, compared with sampling only from the top examples. These evaluations were done using the 131k latent SAE trained on the residual stream of Gemma 2 9b. The scorer and the explainer model were Llama 3.1b 70b instruct, quantized to 4 bits.

### A.6.5  DEPENDENCE ON THE NUMBER OF EXAMPLES

The number of examples shown to the model seems to saturate, at least with the explainer model we used. These evaluations were done using the 131k latent SAE trained on the residual stream of Gemma 2 9b. The scorer and the explainer model where Llama 3.1b 70b instruct, quantized to 4bit.

### A.6.6  EXPLAINABILITY ACROSS LAYERS

We find that, in the case of the 131k latent SAE trained on the residual stream, the earliest layers have lower scores than the later layers. These evaluations were done using the 131k latent SAE trained on the residual stream of Gemma 2 9b. The scorer and the explainer model where Llama 3.1b 70b instruct, quantized to 4bit.

Table A6: Impact of number of examples on Fuzzing and Detection performance. Numbers shown are the median score and the interquartile (25%-75%) range.

| Experiment | Fuzzing | Detection | Embedding |
|---|---|---|---|
| Random interpretation | 0.51 (0.45–0.57) | 0.51 (0.45–0.58) | 0.51 (0.44–0.57) |
| Randomly initialized Topk SAE | 0.55 (0.50–0.60) | 0.54 (0.50–0.59) | – |
| Shown 10 examples | 0.73 (0.62–0.85) | 0.71 (0.58–0.82) | 0.64 (0.54-0.74) |
| Shown 20 examples | 0.74 (0.64–0.85) | 0.72 (0.60–0.84) | 0.66 (0.54-0.76) |
| Shown 40 examples | **0.76 (0.67–0.86)** | **0.74 (0.63–0.85)** | **0.68 (0.57–0.80)** |
| Shown 60 examples | 0.75 (0.66–0.85) | 0.73 (0.62–0.84) | 0.68 (0.57–0.79) |

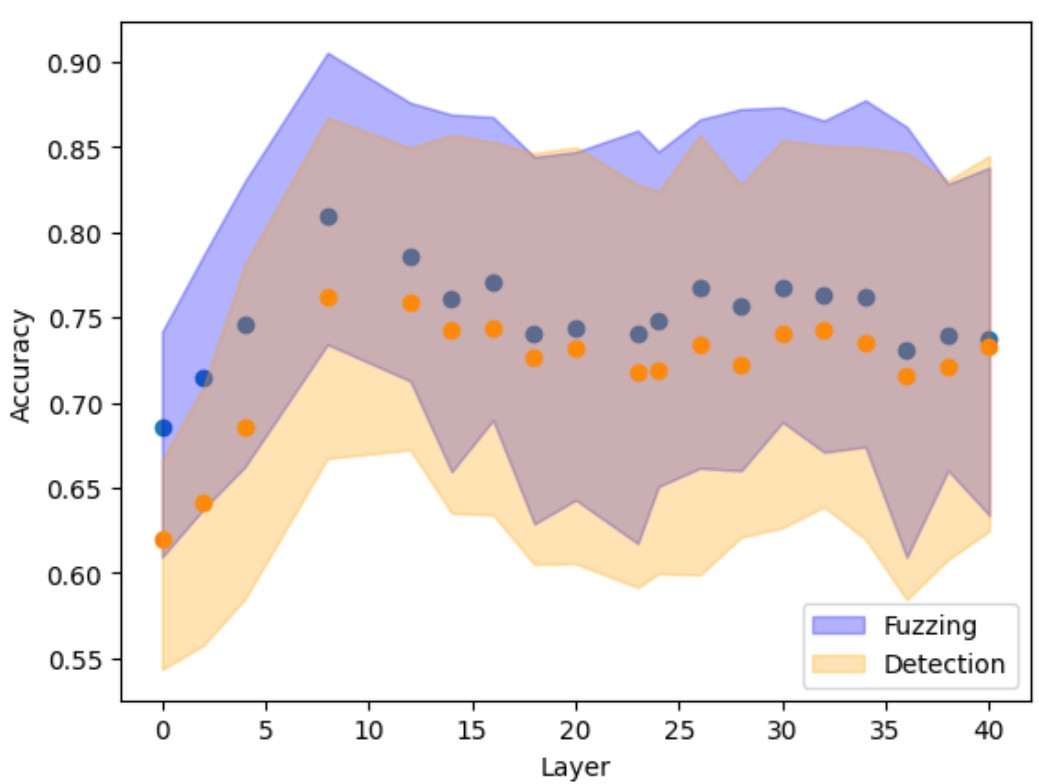

Figure A5: **Accuracy on fuzzing and detection scoring** Dots correspond to the median score over c.a. 300 latent interpretations, and colored region denotes the interquartile range.

Table A7: Comparison of scores of interpretations given by different models. Numbers shown are the median score and the interquartile (25%-75%) range.

| Experiment | Fuzzing | Detection | Embedding | Simulation |
|---|---|---|---|---|
| Random interpretation | 0.51 (0.45–0.57) | 0.51 (0.45–0.58) | 0.51 (0.44–0.57) | -0.02 (-0.02–0.00) |
| Claude | 0.75 (0.68–0.84) | **0.75 (0.65–0.85)** | 0.70 (0.58–0.81) | 0.30 (0.28–0.32) |
| Llama 70b | **0.76 (0.67–0.86)** | 0.74 (0.63–0.85) | 0.67 (0.57–0.80) | 0.29 (0.28-0.32) |
| Llama 8b | 0.70 (0.60–0.81) | 0.70 (0.59–0.81) | 0.64 (0.54–0.75) | 0.26 (0.23-0.30) |
| Human | 0.75 (0.66–0.85) | 0.74 (0.64–0.85) | **0.71 (0.62–0.81)** | **0.36 (0.32–0.39)** |

Table A8: Comparison of scorer models for fuzzing, detection and simulation. Numbers shown are the median score and the interquartile (25%-75%) range.

| Experiment | Fuzzing | Detection | Simulation |
|---|---|---|---|
| Claude | **0.76 (0.66–0.88)** | 0.72 (0.58–0.87) | 0.33 (0.29–0.36) |
| Llama 70b | 0.76 (0.67–0.86) | **0.74 (0.63–0.85)** | 0.29 (0.28–0.32) |
| Llama 8b | 0.70 (0.60–0.81) | 0.70 (0.59–0.81) | 0.26 (0.25–0.31) |

### A.6.7 DEPENDENCE ON SIZE OF EXPLAINER MODELS

Having a larger explainer model leads to better interpretations, but our results suggest that this benefit tends to saturate, as we find that the scores of explanations given by Claude or Llama 70b are very similar. We only did human interpretation of 100 latents. Simulations scores are also only computed over 100 latents.

### A.6.8 DEPENDENCE ON SIZE OF EXPLAINER MODELS

We find that both fuzzing, detection and simulation scoring are dependent on the size of the explainer model, that is, given the same explanations, larger models achieve higher scores on average. While Claude and Llama 80b have similar fuzzing and detection scores, simulation, being more complex, is better performed by Claude.

### A.6.9 DEPENDENCE ON SAE SIZE AND LOCATION

We compare the interpretability of SAEs with different number of latents, trained on the same model, with the neurons of that model. We find that the SAE with the highest number of latents to have the highest scores in the case of the Gemma 2 9b SAEs and that SAEs trained on the residual stream have higher scores for both Gemma 2 9b and Llama 3.1 8b.

The scorer and the explainer model were Llama 3.1 70b instruct, quantized to 4bit.

## A.7 AUTOMATICALLY INTERPRETING INTERVENTIONS

### A.7.1 SCORING IMPLEMENTATION

For each latent, we sample a pool of 40 length-64 prompts from RPJv2 (Computer, 2023), of which 30 i.i.d. prompts are taken for scoring, while the remaining 10 are used by the explainer. The pool is sampled among nonzero activating contexts, stratified by the quintile of the latent's max activation in that context (i.e., the nonzero activating documents are sorted by the context's maximum activation of the current latent, then 8 contexts are sampled from the first quintile, 8 from the second etc.). Each of these prompts is then truncated to include only the first token on which the latent activates, so that all previous activations for the latent are 0. Activations on the first token position are ignored because the SAEs were not trained on that position.

We filter out latents that activates on less than 200 of the 10 million RPJ-v2 tokens.

We then sample generations with a maximum of 8 new tokens from the subject model. For generations with the intervention, we perform an additive intervention on the latent at all token positions

Table A9: Comparison of SAEs for Fuzzing and Detection. Numbers shown are the median score and the interquartile (25%-75%) range.

| Experiment | Fuzzing | Detection |
|---|---|---|
| Random interpretation | 0.51 (0.45–0.57) | 0.51 (0.45–0.58) |
| Randomly initialized Topk SAE | 0.55 (0.50–0.60) | 0.54 (0.50–0.59) |
| 131k latents Gemma 2 9b | **0.76 (0.67–0.86)** | **0.74 (0.63–0.85)** |
| 16k latents Gemma 2 9b | 0.73 (0.63–0.83) | 0.70 (0.59–0.79) |
| Top 32 neurons Gemma 2 9b | 0.62 (0.54–0.70) | 0.59 (0.53–0.64) |
| Top 256 neurons Gemma 2 9b | 0.59 (0.53–0.65) | 0.57 (0.52–0.62) |
| 262k latents Llama 3.1 8b MLP | 0.79 (0.69–0.86) | 0.79 (0.64–0.85) |
| 262k latents Llama 3.1 8b | **0.81 (0.71–0.86)** | **0.83 (0.68–0.85)** |
| Top 32 neurons Llama 3.1 8b | 0.55 (0.51–0.62) | 0.53 (0.50–0.59) |
| Top 256 neurons Llama 3.1 8b | 0.54 (0.49–0.60) | 0.53 (0.50–0.57) |

after and including the final prompt token. We tune the intervention strength of each latent to various KL-divergence values on the scoring set with a binary search. We stop the binary search when the KL divergence is within 10% of the desired value[3]. For our zero-ablation experiment, instead of doing an additive intervention we clamp the latent's activation to 0. Specifically, the hidden states are encoded by the SAE, then the SAE reconstruction error is computed using a clean decoding, then the SAE encoding is clamped and decoded, and the error is added to the clamped decoding.

The scorer is Llama-3.1 8B base. We use the base model for improved calibration, and prompt it as follows.

```
<PASSAGE>
from west to east, the westmost of the seven wonders of the
world is the great wall of china

The above passage contains an amplified amount of "Asia"

<PASSAGE>
Given 4x is less than 10, 4

The above passage contains an amplified amount of "numbers"

<PASSAGE>
In information theory, the information content, self-
information, surprisal, or Shannon information is a basic
quantity derived by her when she was a student at Windsor

The above passage contains an amplified amount of
"she/her pronouns"

<PASSAGE>
My favorite food is oranges

The above passage contains an amplified amount of
"fruits and vegetables"

<PASSAGE>
...
```

---

[3]Sometimes the KL-divergence is not perfectly monotonic in the intervention strength so 10% error is exceeded. We report the average KL divergence that we observe in Figure A6

### A.7.2 GENERATING INTERPRETATIONS

We generate interpretations using only the intervention's effect on the subject's next-token probabilities because this leads to a concise and precise prompt. The explainer, like the scorer, is Llama-3.1 8B.

The explainer sees a distribution of 10 prompts that is sampled i.i.d. from the same population as the scorer's prompts.

We use the following prompt for the explainer, with 3 few-shot examples truncated for brevity.

```
We're studying neurons in a transformer model. We want to know
how intervening on them affects the model's output.

For each neuron, we'll show you a few prompts where we
intervened on that neuron at the final token position, and the
tokens whose logits increased the most.

The tokens are shown in descending order of their probability
increase, given in parentheses. Your job is to give a short
summary of what outputs the neuron promotes.

Neuron 1
<PROMPT>Given 4x is less than 10,</PROMPT>
Most increased tokens: ' 4' (+0.11), ' 10' (+0.04),
' 40' (+0.02), ' 2' (+0.01)

<PROMPT>For some reason</PROMPT>
Most increased tokens: ' one' (+0.14), ' 1' (+0.01),
' fr' (+0.01)

<PROMPT>insurance does not cover claims for accounts
with</PROMPT>
Most increased tokens: ' one' (+0.1), ' more' (+0.02),
' 10' (+0.01)

interpretation: numbers

Neuron 2
...
```

### A.7.3 BASELINES

**Random SAE**. We experiment with a random top-k SAE with $k = 50$, roughly the average sparsity of the gemma SAEs. The encoder is initialized with 131,072 spherically uniform unit-norm latents, and the decoder is initialized to its transpose. We use a random SAE with TopK activations because we need sparsity for our sampling procedure to work properly.

**Random interpretations**. For each layer and target KL value, we compute a random interpretation baseline where we shuffle the interpretations across latents.

### A.7.4 INTERVENTION INTERPRETABILITY RESULTS

Figure A6 shows the average intervention score at each layer, for a few KL values. Intervention scores are higher in later layers, likely because of the increased proximity to the model's output.

Figure A7 is similar to the right half of Figure 4, but at multiple layers and KL values. Early layers have small intervention scores across the board, so there is little correlation, while the reason for little correlation in layer 41 is less clear.

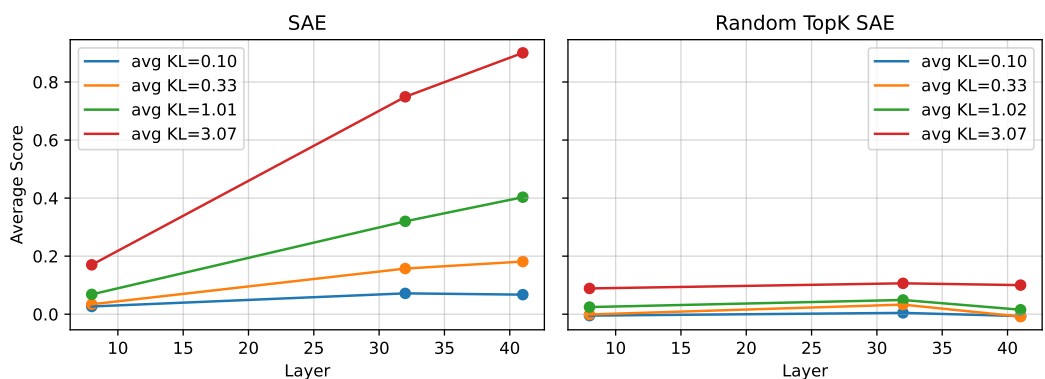

Figure A6: Average intervention score vs layer. SAE latents in later layers have more explainable effects on output. Random latents, however, have uninterpretable effects on output, even at late layers.

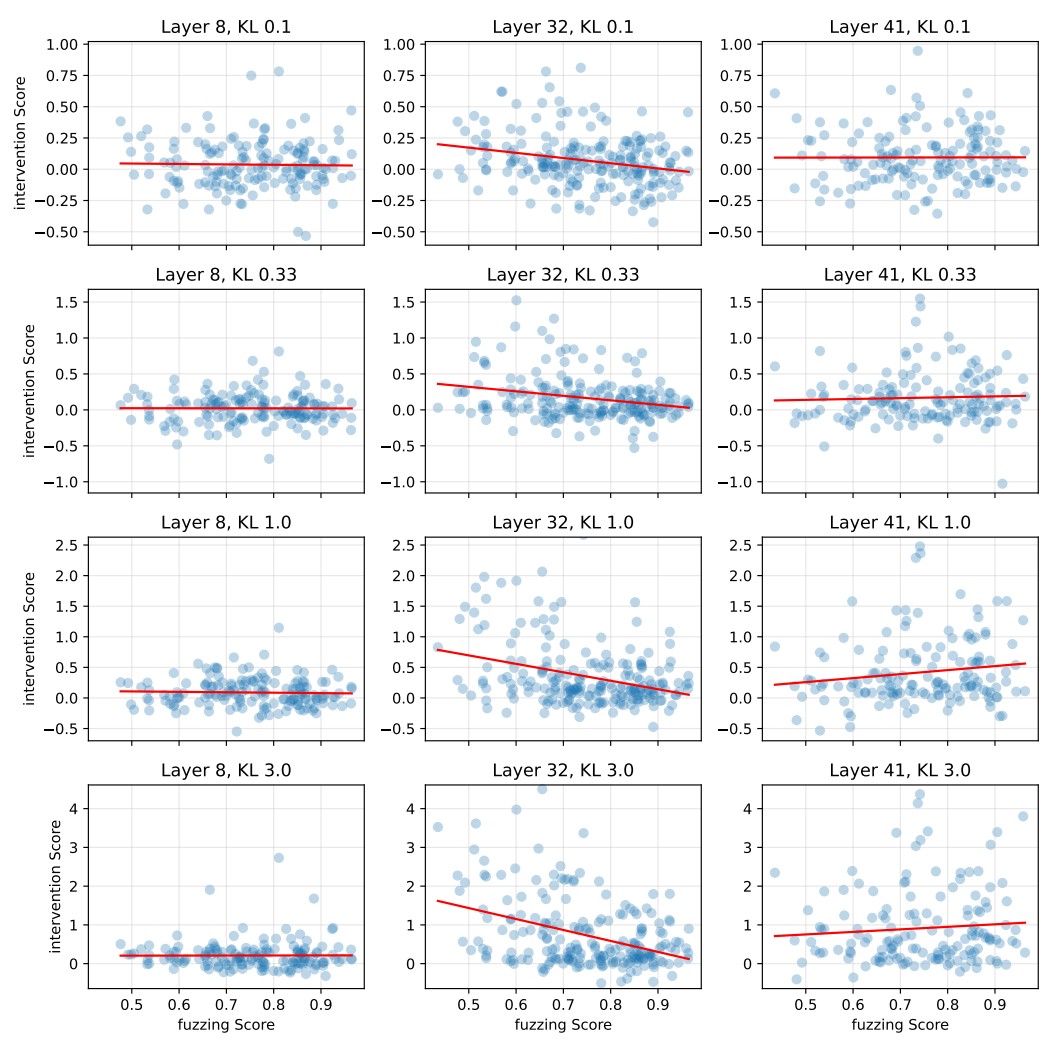

Figure A7: Comparison of intervention and fuzzing scores across layers at various target KL values.

