# OpenReview forum: "Automatically Interpreting Millions of  Features in Large Language Models"
_ICLR.cc/2025/Conference — Submitted to ICLR 2025_

### Official Review · Reviewer_4Nph · 2024-10-21

**Soundness:** 2
**Presentation:** 1
**Contribution:** 2
**Rating:** 3
**Confidence:** 3

**Summary:**

The paper presents an open-source automated pipeline that uses large language models to generate natural language explanations for millions of features in Sparse Autoencoders (SAEs), addressing the challenge of manual interpretation. It introduces five efficient scoring techniques, including intervention scoring, and demonstrates that SAEs are more interpretable than neurons, offering insights into semantic similarity across model layers.

**Strengths:**

- The scale gets improved from previous sota.
- This work lies in an interesting direction.

**Weaknesses:**

- This work did not compare the new evaluation metric with the previous evaluation metric (https://openai.com/index/language-models-can-explain-neurons-in-language-models/) in a solid form. Having similar conclusions as previous approach should not serve as a solid evidence that the new metric is as good as the previous evaluation metric.
- Comparing SAEs in adjacent layers (1) lacks support of motivation and (2) is not well supported. See Questions for details.
- The presentation is poor: readers can not capture the main contribution of this paper with a normal reading flow. The contribution of this work seems to be concentrated on a new evaluation metric, but Section 5 cuts in to discuss about behaviors of the behaviors of SAEs. I would strongly recommend to reorganize the paper to one single claim, with evidence from both sides supporting it.

**Questions:**

1. How is the correlation of the proposed evaluation metric with the original metric?
2. How is the efficiency of the proposed evaluation metric compared to the original metric?
3. Previous work (https://transformer-circuits.pub/2023/monosemantic-features) has already shown SAEs have more interepretable features than neurons. What's the purpose of validating this result?
4. Why comparing SAEs in adjacent layers? (Why is it interesing?)
5. How is comparing SAEs in adjacent layers related to your evaluation metric?
6. Are there any prior work that supports your method in evaluating the SAEs in adjacent layers?

---

> ### Author Response · Authors · 2024-11-21
>
> > This work did not compare the new evaluation metric with the previous evaluation metric (https://openai.com/index/language-models-can-explain-neurons-in-language-models/) in a solid form...
>
> We have extended the analysis of the previous evaluation method, by introducing a cost analysis as well as a deeper discussion on some problems with simulation scoring— see Section 4.1 and Appendix A.4.6. See also discussion with reviewers YaT1, s55e.
>
> > Comparing SAEs in adjacent layers (1) lacks support of motivation and (2) is not well supported. See Questions for details...
>
> We have since decided to remove this section, responding to your and other reviewers’ comments about readability. We agree that it significantly cuts into the narrative of the article and makes it harder to follow. We still find it relevant to reply to some of these points in Question 4 and 5.
>
> **Questions**
> > How is the correlation of the proposed evaluation metric with the original metric?
>
> We have reported both the Spearman and the Pearson correlation between the original metric and the proposed metrics in the original manuscript. Although only fuzzing has a high correlation with simulation scoring (>0.7), we believe that the added discussion about the drawbacks and advantages of the different methods justifies the introduction of the new scores. See also replies to reviewers YaT1, s55e.
>
> > How is the efficiency of the proposed evaluation metric compared to the original metric?
>
> We have now provided some comparison between the methods to support the claim of efficiency.  We compare the number of input and output tokens used for each of the explanation methods with cost estimates at different explainer model sizes— see new Table 1 and lines 351-359. Simulation is about 5 times more expensive than detection and fuzzing if one is able to run the scorer model locally, but closer to 30 times more expensive when using a large closed source model, because providers do not allow for the tools required to make the method more efficient. Embedding on the other hand is 50 times cheaper than simulation, even when compared to the price of running it locally.
>
> > Previous work (https://transformer-circuits.pub/2023/monosemantic-features) has already shown SAEs have more interepretable features than neurons. What's the purpose of validating this result?
>
> We have decided to include the neuron comparison as there is still significant effort in [improving explanations of neurons](https://transluce.org/neuron-descriptions), and we wanted to evaluate how effective our pipeline would be at finding and scoring explanations of neurons. Due to the fact that the exact pipeline used by Anthropic is not known, we decided that it made sense to reproduce their findings on neurons, using our open-source pipeline. We were also interested in “sparsifying” neurons with the top-k function as a way to improve their interpretability, and so decided that it was important to include negative results on sparsified neurons.
>
> > Why comparing SAEs in adjacent layers? (Why is it interesing?)
>
> Recent work has shown that training a single SAE on all layers is effective (Lawson 2024). Our work adds to this discussion, suggesting that residual stream SAEs on adjacent layers are learning very similar latents, and are probably a waste of compute. SAEs trained on the MLP output of adjacent layers probably don’t have the same problem, and so should be prioritized.
>
> > How is comparing SAEs in adjacent layers related to your evaluation metric?
>
> Although a significant portion of our work relates to the results of the different scoring techniques, a great deal of effort was put into generating explanations for all latents of several SAEs. This gave us a legitimate opportunity to compare explanations of consecutive layers and estimate their overlap.
>
> > Are there any prior work that supports your method in evaluating the SAEs in adjacent layers?
>
> We are not evaluating SAEs per se, but only measuring the alignment of their decoder matrices. Using the Hungarian algorithm to align and compare independently trained networks has been done before ([Ainsworth 2022](https://arxiv.org/abs/2209.04836)).

---

### Official Review · Reviewer_s55e · 2024-11-03

**Soundness:** 2
**Presentation:** 2
**Contribution:** 3
**Rating:** 6
**Confidence:** 4

**Summary:**

This paper investigates sparse autoencoders (SAEs), which project activation representations into a sparse high-dimensional latent space for interpretability. To automatically explain the large number of latent features, this paper proposes an LLM-based framework, explaining millions of latents across multiple models, layers, and SAE architectures. Four evaluation methods are proposed, including detection, fuzzing, surprisal, embedding, measuring the extent to which an explanation enables a scorer to discriminate between activating and non-activating contexts. Additionally, an intervention scoring is proposed to interpret a feature’s counterfactual impact on model output.

**Strengths:**

-	This paper focuses on an important research problem of SAEs producing a large number of latent features that require automatic explanations and evaluations.
-	Several evaluation metrics have been proposed to assess the generated explanations, comparing them across different explainer models, SAEs, and layers.
-	Some findings are interesting. For instance, sampling examples that are shown to the explainer model may increase the scores of features, highlighting a problem with current auto-interpretability evaluations. Experimental results suggest a priority for training wider SAEs on a smaller subset of residual stream layers. These may provide valuable insights for future research.

**Weaknesses:**

-	The writing and clarity of the paper could be improved. The current version is somewhat difficult to follow and understand. Please see my detailed questions below.
-	Many observations are presented without in-depth analysis or further exploration. For instance, in Section 3.1, the impact of context length and latent space size on activation data is mentioned, but it is unclear how these results relate to the selection of 256 tokens. In Section 3.2, the claim that “showing such short contexts to the explainer model hinders the correct identification of latents with complex activation patterns” seems to contradict the use of short activating examples with 32 tokens. In Section 3.3, while qualitative results for different sampling strategies are provided, it is not clear how to optimize the sampling process. Additionally, in Section 4.1, the statement “The imperfect correlations hint at either shortcomings of the scoring metrics or the fact that these metrics can measure different qualities of explanations” lacks clarity. What specific qualities of explanations are measured by these automatic metrics?
-	The generated explanations largely rely on the prompts and explainer models. It is unclear how generalizable the results are and to what extent specific prompts and explainer models may affect the quality of the generated explanations.
-	While several automatic evaluations are compared, it is unclear whether they accurately reflect the quality of explanations. It would be beneficial to conduct human evaluations, at least on a small set, and compare the correlation with automatic evaluations.

**Questions:**

-	Why is the context length of 256 chosen for activation collection? Is this value based on empirical selection?
-	Why is the activating example limited to only 32 tokens, given that short contexts may hinder the correct identification of latents with complex activation patterns?
-	What accounts for the differences in explanations between “randomly sampling” and “uniformly sampling”?
-	In Section 5, the statement that “if the explanation for latent α at layer j is very different from the explanation for the same latent at layer j + 1, this would suggest that our pipeline is inconsistent and noisy” raises a hypothesis. Can you elaborate on it?

---

> ### Author Response · Authors · 2024-11-21
> **Response to Weaknesses**
>
> > The writing and clarity of the paper could be improved...
>
> We are increasing the clarity of the paper by moving the section comparing SAEs at different layers to the Appendix to allow for more discussion space for readers who are not familiar with SAEs, as well as to address your questions and comments below.
>
> > Many observations are presented without in-depth analysis or further exploration...
>
> We agree with the reviewer that more in-depth discussion should be provided in some sections, which was not originally done due to space limitations. We reply to the duplicate questions in the question sections, and to the others here.
>
> In Section 3.2, lines 160-165, we include a discussion on whether practitioners should care more about “top” explanations or “stratified” explanations. We argue that we should care about more than just the top activating examples. This means that there is not an optimal sampling process, because it depends on the type of explanation one is looking for. We believe that this responds to the comment: “ while qualitative results for different sampling strategies are provided, it is not clear how to optimize the sampling process.”.
>
> We agree that the claim in Section 4.1 is vague, and have significantly reworded it (lines 374-391). Different scoring methods “target” different qualities of explanations:
>
> For example, consider a latent whose automatically generated interpretation is "activates on legal documents" without a reference to specific activating tokens. If the actual latent activates on various unrelated words in legal documents, this interpretation would score highly on detection, embedding, and surprisal. However, fuzzing and simulation would yield lower scores since they require a scorer model to predict specific tokens.
>
> Consider an alternative interpretation of that latent: "activates on the token <_of>". Detection, embedding, and surprisal would classify the token as activating on a broad array of documents, yielding many false positives and a low score. Fuzzing and simulation could easily pick a single token and score highly, mostly because current scoring techniques limit the number of shown examples.
>
> That’s why we are proposing a “battery” of tests, that can be used to sort explanations and investigate whether the explanation can be improved or whether the latent is not easily interpretable. A more complete explanation, “activates on the token <_of> in legal documents” would score high in all scoring methods and could be more trusted. We have added an Appendix section, A.4.7, where we show examples of explanations whose scores disagree.
>
> > The generated explanations largely rely on the prompts and explainer models...
>
> We agree that the generated explanations depend significantly on the prompts and explainer models. In the future we would like to continue optimizing the prompts, although we already did a considerable amount of prompt tuning in preliminary experiments. We already show in this paper how the quality of the explanations depends on the model.
>
> We compare the scores of explanations generated with Llama 8b, 70b and Claude Sonnet 3.5, using detection, fuzzing, embedding and simulation scoring, and show that larger models improve the explanations generated, but that the improvement from 8b to 70b is larger than from Llama 70b to Sonnet 3.5.
>
> We also tested the effect of chain-of-thought and adding different information to the prompt, as discussed in lines 483-511 and in Appendix A5, namely A.5.2, A.5.3, A.5.5 and A.5.7.
>
> > While several automatic evaluations are compared, it is unclear whether they accurately reflect the quality of explanations. It would be beneficial to conduct human evaluations, at least on a small set, and compare the correlation with automatic evaluations.
>
> We agree with the reviewer that scores should accurately reflect the quality of explanations. We have done human scoring and found the correlations between these scores and all the proposed methods. For human scoring, we used a rubric score similar to that used by [Anthropic](https://transformer-circuits.pub/2024/scaling-monosemanticity/index.html), see details in Appendix A.4.6, finding that the detection score, fuzzing score, and simulation score are all correlated with human scores at around 0.60, but other methods are less correlated with human judgment (<0.40).
>
> However, the human scores may not be more valid than the automatic evaluations. For each latent explanation, the human saw 10 contexts on average, whereas the automatic methods saw at least 100 activating contexts, and fuzzing, detection, embedding and surprisal saw another 100 non-activating contexts. This difference in scale is one of the main reasons we believe automatically generated interpretations and evaluations truly shine.

---

> ### Author Response · Authors · 2024-11-21
> **Response to Questions**
>
> > Why is the context length of 256 chosen for activation collection? Is this value based on empirical selection?
>
> When we originally started the project, we selected 256 tokens as an intermediate value between the context length that the SAEs were trained with (1024 tokens) and our planned context length for generating explanations – either 32 or 64. We wanted flexibility to explore larger context lengths if given the opportunity. Because we were already expecting latents whose explanations required larger contexts to have lower scores, we were not concerned about leaving them out of our pipeline.
>
> > Why is the activating example limited to only 32 tokens, given that short contexts may hinder the correct identification of latents with complex activation patterns?
>
> In Section 3.2 we point to the fact that there may be more complex latents that our methods do not completely capture, and we expect those to have low scores. We don’t think it makes sense to use larger context for all latents since our results imply that a large fraction of latents is already explained with lengths 32 or 64. Using larger contexts would make the whole pipeline more expensive with low return.  We are also able to sort latents that can be easily explained with the current context length, from those that have low scores and could potentially require larger context lengths to explain, and have included a discussion in the text addressing these points, lines 160-165.
>
> > What accounts for the differences in explanations between “randomly sampling” and “uniformly sampling”?
>
> Thank you for this question, which we agree is not clear in the paper. What we meant by “uniform sampling” was stratified sampling, where we divide the data into ten bins and sample a fixed number of samples from each bin. We have corrected the labels in the figures and in the main text.
>
> We used stratified sampling to guarantee that there are samples in each part of the activation distribution, due to the fact that we choose only 40 examples for each latent. We expect these two distributions to be very similar, and in fact that is what we empirically observed.
>
> > In Section 5, the statement that “if the explanation for latent α at layer j is very different from the explanation for the same latent at layer j + 1, this would suggest that our pipeline is inconsistent and noisy” raises a hypothesis. Can you elaborate on it?
>
> We have now removed this section due to the multiple comments related to improving the readability of the article, but we will answer your question regardless. We were arguing that consecutive SAEs trained on the residual stream would find the same “latent” in both layer i and layer j.
>
> This latent would obviously not have the same index, as discussed in the previous version of the manuscript, but after the proposed alignment technique we could find the latents a and a’ at layer i and layer j respectively that were the most aligned.
>
> If the alignment were substantial, we would expect that their explanations would be similar, because they would have the same effect in the residual stream. This argument ignores the fact that we are aligning the decoder directions but not the encoder directions, so the activating contexts could be different.
>
> The encoder and decoder directions are initialized to be identical, so we expect them to generally be similar, but it is possible for them to drift apart during training. If it were the case that many features had similar decoder directions but very different encoder directions and so had different explanations, it would point to a weakness in our Section 5 experiments.

---

> > ### Comment · Reviewer_s55e · 2024-11-27
> >
> > Thank you for answering my questions, providing clarifications, and conducting additional experiments, including a human evaluation. Most of my concerns have been addressed, so I’ve decided to increase my score.

---

### Official Review · Reviewer_8Dae · 2024-11-04

**Soundness:** 4
**Presentation:** 3
**Contribution:** 4
**Rating:** 8
**Confidence:** 4

**Summary:**

The paper builds an open-source automated pipeline to generate and evaluate natural language explanations for sparse autoencoder features with LLMs. The framework has been evaluated on various dimensions, including SAE size, activation function, loss, and LLMs. Five new scoring techniques are proposed. The paper finds that SAEs trained on nearby layers' the residual stream are highly similar. And they are also more interpretable than neurons.

**Strengths:**

- The open-source framework is comprehensive and valuable for large-scale SAE analysis. The experiments are well designed to illustrate the effectiveness of the proposed method.
- The metrics proposed in this paper provides more dimensions to evaluate the generated explanations, which would be valuable for the SAE community.
- Some of the findings are meaningful to the SAE community. For example, larger latent SAE learn more dataset-specific latents. The relations between different sampling approaches and the generated explanations. And the high correlations between latents at adjacent layers.

**Weaknesses:**

- The method is a bit hard to understand for readers who are not familiar with SAEs. For example, how section 3.1 is related to 3.2? Would be more illustrative if a figure of the whole pipeline is provided.
- It would be more clear to provide a simple example of explaining the latents of SAEs. And even better if an example involves the whole workflow of this framework is provided.

**Questions:**

The framework proposed in this paper is a valuable tool for the SAE community. It provides an automated pipeline to generate and evaluate natural language explanations for sparse autoencoder features. However, the authors better consider to write a more accessible version for readers who are not familiar with SAEs, especially for section 3. It is a bit difficult to follow it.

---

> ### Author Response · Authors · 2024-11-21
>
> > The method is a bit hard to understand for readers who are not familiar with SAEs...
>
> We thank the reviewer for this excellent suggestion. Figure 1 now illustrates the full pipeline, from collecting activations, to generating interpretations and scores.
>
> > ...However, the authors better consider to write a more accessible version for readers who are not familiar with SAEs, especially for section 3. It is a bit difficult to follow it.
>
> Taking into account the comments of several reviewers, we are increasing the accessibility of the article. We are moving the section comparing SAEs at different layers to the Appendix, to allow for more discussion space in the body for readers who are not familiar with SAEs, (see e.g. lines 93-97). We also made changes to Sections 3.1, 3.2 and 3.3, expanding the discussion on each scoring method.

---

### Official Review · Reviewer_YaT1 · 2024-11-04

**Soundness:** 2
**Presentation:** 3
**Contribution:** 3
**Rating:** 5
**Confidence:** 4

**Summary:**

This paper introduced five automated scoring methods to score the explanations of SAE latents, and discussed the shortcomings of existing scoring techniques. The paper also finds that SAE trained on nearby layers are highly similar, and provided actionable insights for practitioners to train wider SAEs instead of narrower SAEs to be efficient when there’s a compute constraint.

**Strengths:**

- Given the large sizes of SAEs nowadays and an increasing need for model explainability, automatically generating explanations of SAE latents efficiently is an important topic.
- The paper is well written, with ablations of design choices clearly described.

**Weaknesses:**

- Format: There are no line numbers, and it's showing "Under review as a conference paper at ICLR **2024**" instead of **2025** at the top.
- The low correlation between different evaluation methods in Table 1 is concerning. Since the simulation method proposed in prior work is vetted and established, the new ones proposed in this work should at least have strong rank correlation (> 0.7) with it to prove that they work. Since this the scoring methods are the primary contribution in this paper, the authors should conduct more rigorous tests to ensure their validity. I would also encourage the authors to conduct experiments with ground truth explanations (e.g., SAE latents with known explanations found in prior work, or easily constructed an embedding model responding to a known concept/explanation), to make a stronger case in terms of the reliability of these new methods.
To add on, instead of correlation of the raw scores, it might make more sense to look at the "rank" correlations of different methods.
- The authors claim that the new methods are more efficient than prior scoring methods without actually quantifying the efficiency gain to support the claim. If efficiency gain is one of the highlights of these scoring methods, the authors should consider comparing runtime of different methods to support the claim.
- The author mention “Our large-scale analysis confirms that SAE latents are indeed much more interpretable than neurons, even when neurons are sparsified using top-k postprocessing.” in the abstract as one of the main findings, but the details cannot be found in the main paper but in the appendix. The authors should consider moving it to the main paper if this is one of the main claims.
- Reproducibility: The author mentioned a plan to open-source the project, but it's hard to evaluate the quality of their code for reproducibility purpose either since it's not provided as one of the supplementary files.

**Questions:**

- Missing a highly relevant work, "Explaining black box text modules in natural language with language models". How does their scoring method compare to the ones proposed in this paper?
- Can the authors explain  the negative correlation between the fuzzing score and intervention score in figure 4? If they are both useful scoring methods, why would the correlation be negative?
- Unlike the claim in the paper, figure 5 is still showing statistical alignment across layers. Can the authors provide evidence for semantic similarity? (e.g., compute explanation similarity across layers instead of the matrix statistics)

*Minor issues*
- The authors say “we introduced *five* new techniques” in the abstract, and “We addressed issues with the conventional simulation-based scoring and introduced *four* new scoring techniques” in the conclusions. The readers might get confused in terms of the number of methods actually introduced in the paper, if it’s five, please be consistent throughout the paper.

---

> ### Author Response · Authors · 2024-11-21
>
> > The low correlation between different evaluation methods in Table 1 is concerning...
>
> We respectfully disagree with the claim that simulation scoring is vetted and established. Prior work has found that explanations with high simulation scores can have high error rates in intervention tasks, and a low F1 score on a task similar to detection, see [Huang (2023)](https://arxiv.org/pdf/2309.10312). In [recent work](https://arxiv.org/pdf/2406.04093), the OpenAI team also discussed downsides of simulation scoring. Simulation focuses on the activating part of the distribution and asks whether an explanation allows us to predict how strongly a latent is active, while we are also interested in distinguishing active from non-active latents, which is not directly measured by simulation scoring.
>
> While our scores are correlated with simulation scoring, they are fundamentally measuring different quantities which are just as important as what simulation measures. As an added benefit, our methods are also cheaper and more efficient to run. We discuss in Section 4.1 some hypotheses on why there is a weak correlation between our methods and simulation.
>
> We agree it would be very useful to compare our explanations to ground truth explanations, but we are not aware of a big enough dataset of such explanations. In lieu of this, we do evaluate human-generated explanations, and provide human generated scores for automatically generated explanations. We find that human-generated explanations have similar fuzzing, detection, and embedding scores to automatically generated explanations, while having slightly higher simulation scores. We find that human scores have a correlation of around 0.6, to detection, fuzzing and simulation; for details see the response to review 8Dae, as well as Appendix A.4.
>
> > The authors claim that the new methods are more efficient than prior scoring methods...
>
> Thank you for this comment. This information was left out as an oversight. We have now added some quantitative comparison between the methods to support our claim that the new methods are more efficient and cost-effective. We compare the number of input and output tokens used for each of the explanation methods with cost estimates at different sizes— see new Table 1 and lines 351-359. Simulation is about 5 times more expensive than detection and fuzzing if one is able to run the scorer model locally, but closer to 30 times more expensive when using a large closed source model, because providers do not allow for the tools required to make the method more efficient. Embedding on the other hand is 50 times cheaper than simulation, even when compared to the price of running it locally.
>
> > Reproducibility: The author mentioned a plan to open-source the project, but it's hard to evaluate the quality of their code for reproducibility purpose either since it's not provided as one of the supplementary files.
>
> We thank the reviewer for this suggestion and apologize for the oversight. We uploaded our code to an anonymous repository: https://anonymous.4open.science/r/interpreting_latents/, which comes with examples and a guide on how to use the library.
>
> > Missing a highly relevant work, "Explaining black box text modules in natural language with language models". How does their scoring method compare to the ones proposed in this paper?
>
> We already mentioned a scoring technique that is based on computing the activations of a model on generated examples [(Koft 2024)](https://arxiv.org/abs/2405.20331), but we were unaware that there was an earlier example of using that technique on language models. We have now added a reference to "Explaining black box text modules in natural language with language models" as it is a relevant work.
>
> Their explanation generation method is similar to ours, although they compute activations on trigrams, while we compute the activations on a dataset that has a similar distribution to the training dataset. The fact that this type of scoring requires two distinct steps, one where an LLM generates trial contexts, and another where the contexts are run through a model and the SAE and the activation collected, leads us to not focus on “generative” scoring in this article, although we believe it could have its merits.
>
> > Can the authors explain the negative correlation between the fuzzing score and intervention score in figure 4? If they are both useful scoring methods, why would the correlation be negative?
>
> We argue that there are some latents that are better explained by their effects on the outputs of the model instead of on their activation patterns. Given this hypothesis, it makes sense that there would be a negative correlation between these two scores. In Figure 4, we show that latents with low “correlational” (fuzzing) scores are likely to be better explained by their “counterfactual” (intervention) effect. We have added discussion of this issue on lines 405-417.

---

> ### Author Response · Authors · 2024-11-21
>
> > Unlike the claim in the paper, figure 5 is still showing statistical alignment across layers. Can the authors provide evidence for semantic similarity? (e.g., compute explanation similarity across layers instead of the matrix statistics)
>
> We appreciate this question by the reviewer. To streamline the article and in line with comments from other reviewers, we have taken this section out, as it clashed with the flow of the manuscript. To answer your question, in Figure 5 we are already showing the explanation similarity between layers, and that’s the statistical alignment the reviewer is referring to, instead, in Figure A5 of the appendix we show that the alignment between the decoder elements is non-existent.
>
> **Small changes:**
> We have also corrected the ICLR date, added page numbers and fixed the inconsistency between 4 and 5 different methods. Now, both abstract and conclusion mention 5 methods.

---

> > ### Comment · Reviewer_YaT1 · 2024-12-03
> >
> > Thanks the authors for the response which have helped address most of my questions, and I've adjusted my review accordingly. Given the different focuses of the scoring methods, presenting the correlation metric between methods in the main text might be confusing to the readers due to the low scores. It might be better to move the correlation scores with human explanations to the main text instead.

---

### Meta-Review · Area_Chair_WQ4K · 2024-12-20

**Metareview:**

## Summary of Scientific Claims and Findings
The paper introduces a novel automated framework using large language models (LLMs) to generate and evaluate natural language explanations for Sparse Autoencoder (SAE) features. It proposes five new scoring methods—detection, fuzzing, surprisal, embedding, and intervention scoring—to assess the interpretability of these features. The paper also provides insights into the high similarity of SAEs trained on nearby layers and suggests that wider SAEs are more efficient for certain tasks under computational constraints. The authors claim their methods are more efficient and offer actionable insights for practitioners in model explainability.

## Strengths
1. **Relevance and Importance**: The paper addresses a timely issue of model interpretability in the context of large SAEs, which is crucial given the growing complexity of modern neural networks.
2. **Comprehensive Framework**: The work presents a comprehensive open-source pipeline that could significantly aid researchers in analyzing SAEs.
3. **Novel Scoring Techniques**: The introduction of multiple scoring techniques offers a broader set of tools for evaluating the quality of explanations, contributing valuable methodologies to the community.
4. **Well-Designed Experiments**: The experiments are well-structured to validate the effectiveness of the proposed methods, providing meaningful insights into SAE behavior.

## Weaknesses
1. **Clarity and Presentation**: Several reviewers noted that the paper's presentation could be improved. The organization sometimes makes it difficult to follow the main contributions and arguments.
2. **Validation and Comparison**: The paper lacks a solid comparison with previous evaluation metrics. The correlation between the new and established methods is not robust, which raises questions about the validity and reliability of the new metrics.
3. **Depth of Analysis**: Some findings are presented without extensive analysis, limiting the depth of insights offered by the paper.

 After thorough consideration of other submissions in the same batch, I find myself recommending rejection for this paper. I encourage the authors to refine their work and consider resubmission to a future conference or journal.

**Additional Comments On Reviewer Discussion:**

#### Summary of Discussion and Rebuttal
- **Reviewers' Points**: Reviewers raised concerns about the correlation of new scoring methods with established metrics, the clarity of the presentation, and the lack of human evaluations to validate automatic scoring methods.
- **Authors' Responses**: The authors provided detailed clarifications, added missing comparisons, and uploaded the code to an anonymous repository for reproducibility. They also conducted additional analyses, including human evaluations, to strengthen their claims.
- **Weight in Final Decision**: Several reviewers noted that the paper's presentation could be improved. The validation and comparison part should also be improved. After thorough consideration of other submissions in the same batch, I recommend to reject the paper. I encourage the authors to refine their work and consider resubmission to a future conference or journal.

---

### Decision · Program_Chairs · 2025-01-22

Reject